# Identification of human TERT elements necessary for telomerase recruitment to telomeres

**Jens C Schmidt[†], Andrew B Dalby[†], Thomas R Cech\***

Department of Chemistry and Biochemistry, BioFrontiers Institute, Howard Hughes Medical Institute, University of Colorado Boulder, Boulder, United States

**Abstract** Human chromosomes terminate in telomeres, repetitive DNA sequences bound by the shelterin complex. Shelterin protects chromosome ends, prevents recognition by the DNA damage machinery, and recruits telomerase. A patch of amino acids, termed the TEL-patch, on the OB-fold domain of the shelterin component TPP1 is essential to recruit telomerase to telomeres. In contrast, the site on telomerase that interacts with the TPP1 OB-fold is not well defined. In this study, we identify separation-of-function mutations in the TEN-domain of human telomerase reverse transcriptase (hTERT) that disrupt the interaction of telomerase with TPP1 in vivo and in vitro but have very little effect on the catalytic activity of telomerase. Suppression of a TEN-domain mutation with a compensatory charge-swap mutation in the TEL-patch indicates that their association is direct. Our findings define the interaction interface required for telomerase recruitment to telomeres, an important step towards developing modulators of this interaction as therapeutics for human disease.

**\*For correspondence:** thomas.cech@Colorado.EDU

[†]These authors contributed equally to this work

**Competing interests:** The authors declare that no competing interests exist.

**Reviewing editor**: Carol Greider, Johns Hopkins University, United States

## Introduction

Linear human chromosomes end in G-rich telomeric repeat DNA with a single-stranded overhang (*Blackburn, 2005*). Chromosome ends pose two challenges to cells: they resemble DNA damage sites, and they shorten progressively each cell cycle due to the end-replication problem (*Levy et al., 1992*). The six-protein shelterin complex specifically binds telomeric DNA and prevents the DNA damage machinery from recognizing telomeres (*Palm & de Lange, 2008*). Additionally, shelterin recruits telomerase, an RNA-containing reverse transcriptase that adds telomeric repeats to the 3′-end of the single-stranded overhang to maintain telomere length (*Nandakumar and Cech, 2013*). Loss-of-function mutations in telomerase and shelterin components cause human diseases like dyskeratosis congenita, aplastic anemia, and pulmonary fibrosis (*Armanios and Blackburn, 2012*). In addition, like stem cells and germ cells, 90% of cancers depend on telomerase activity to maintain the potential to proliferate indefinitely (*Stewart and Weinberg, 2006*; *Shay and Wright, 2011*).

Telomerase is a ribonucleoprotein (RNP) complex. Its catalytic core is comprised of hTERT and the telomerase RNA component (hTR), which contains the template used for telomeric DNA synthesis (*Cech, 2004*). In the holoenzyme, hTR also associates with dyskerin for stability and with TCAB1 for telomerase maturation and trafficking to Cajal bodies (*Mitchell et al., 1999*; *Venteicher et al., 2009*). The hTERT protein contains four domains, the telomerase N-terminal domain (TEN-domain), the telomerase RNA-binding domain (TRBD), the reverse transcriptase domain (RT-domain), and the C-terminal extension (CTE) or putative thumb domain of the reverse transcriptase. The TRBD- and RT-domains of hTERT bind hTR and position the template of hTR in the RT-domain active site to synthesize telomeric repeats (*Cech, 2004*). The TEN-domain participates in the catalysis of telomeric repeat addition (*Jacobs et al., 2006*; *Jurczyluk et al., 2011*; *Robart and Collins, 2011*; *Wu and Collins, 2014*). Additionally, both the TEN-domain and CTE contain DAT (dissociates activities of

**eLife digest** In the nucleus of a cell, the DNA that contains the cell's genetic information is packaged into long structures called chromosomes. Every time a cell divides, its chromosomes are duplicated. However, the proteins that are responsible for copying the DNA cannot reach the very end of the DNA strand, causing the chromosomes to progressively shorten. To ensure that this does not cause genetic information to be lost, each chromosome ends in a repetitive stretch of DNA called a telomere. Though the end of the telomere is lost whenever the DNA is copied, an enzyme called telomerase replaces the sequence that has been lost and counteracts the shortening of the telomeres.

Shelterin is a protein complex that binds to telomeres to protect them and also helps telomerase to work correctly. Shelterin contains a specific site that attaches to telomerase, but exactly how the human versions of these two molecules bind to each other is unknown. A possible interaction site had been identified on the telomerase, which, when mutated, stops the telomerase working properly. However, as this region is also involved in lengthening the telomeres after the chromosomes have duplicated, it is not certain that these problems result from the telomerase failing to bind to shelterin.

The enzyme telomerase is unusual; it has both RNA and protein components. Like all other proteins, the telomerase protein is made up of strings of amino acids. Schmidt et al. discovered that replacing two specific amino acids in human telomerase prevents its binding to shelterin. Cells that produced the modified form of the telomerase had chromosomes with shortened telomeres. However, if the cells also produced modified versions of the shelterin complex that were designed to bind to the modified telomerase, telomere length was normal. This indicates that the telomerase interacts directly with shelterin, rather than through a 'bridging' molecule.

Mutations in the genes coding for both shelterin and the telomerase enzyme cause a number of human diseases, and cancers rely on the activity of telomerases to grow. Knowing how shelterin and telomerase interact could therefore help to design drugs that may either restore or disrupt the interaction and therefore can be used to treat these diseases.

telomerase) regions, N-DAT and C-DAT, respectively (**Armbruster et al., 2001**; **Banik et al., 2002**). Mutations in either DAT-region disrupt the ability of telomerase to immortalize cells, but they retain catalytic activity (**Armbruster et al., 2001**; **Banik et al., 2002**). Based on experiments with N-DAT mutants, the TEN-domain has been proposed to mediate telomerase recruitment to telomeres (**Armbruster et al., 2003**, **2004**).

Telomerase is recruited to telomeres in S-phase, while it remains in Cajal bodies for the remainder of the cell cycle (**Jady et al., 2006**; **Tomlinson et al., 2006**). Telomerase localization to Cajal bodies requires the interaction of hTR with TCAB1 (**Venteicher et al., 2009**). TCAB1 is not required for enzymatic activity but for function in vivo (**Venteicher et al., 2009**), indicating that recruitment to Cajal bodies is a key step in telomerase trafficking or maturation. Recruitment of telomerase to the telomere requires the shelterin component TPP1 (**Xin et al., 2007**; **Abreu et al., 2010**). TPP1 is recruited to the shelterin complex via an interaction with TIN2 (**Abreu et al., 2010**). TPP1 also tightly binds to POT1 (**Wang et al., 2007**), which specifically binds the telomeric single-stranded DNA overhang (**Baumann and Cech, 2001**; **Lei et al., 2004**). Biochemical studies have demonstrated that the interaction between the POT1-TPP1 complex and telomerase stimulates its repeat addition processivity (RAP), that is, increases the number of consecutive repeats that telomerase synthesizes in a single round of telomeric DNA addition (**Wang et al., 2007**). Recently, the TEL-patch, a group of mostly acidic amino acids on the oligosaccharide binding (OB)-fold domain of TPP1 was shown to be necessary for telomerase recruitment in vivo and RAP stimulation in vitro (**Nandakumar et al., 2012**; **Sexton et al., 2012**; **Zhong et al., 2012**). Furthermore, cell biological assays demonstrated that the OB-fold domain of TPP1 is sufficient to recruit telomerase to a non-telomeric chromosomal locus (**Zhong et al., 2012**). Although TPP1 is critical for recruitment of telomerase to the telomere, it has been unclear whether TPP1 makes a direct interaction with telomerase or if their interaction is bridged by another component.

Conversely, the site on telomerase that associates with the telomere is not well defined. The TEN-domain of hTERT is a potential candidate to contain key residues necessary for the interaction with

TPP1. A mutation in the TEN-domain of hTERT, G100V, is defective in stimulation of telomerase processivity in vitro (*Zaug et al., 2010*) and fails to localize to telomeres in vivo (*Zhong et al., 2012*). However, G100V telomerase has substantially reduced enzymatic activity (*Zaug et al., 2010*); thus, it is difficult to assess whether its failure to function in vivo is due to an activity or recruitment defect. Furthermore, the interaction between TPP1 and telomerase may require a bridging factor. In *Schizosaccharomyces pombe*, Ccq1 bridges the interaction between Tpz1, the TPP1 homologue, and telomerase (*Miyoshi et al., 2008*; *Moser et al., 2011*). A human homologue of Ccq1 has not been identified (*Nandakumar and Cech, 2013*). Thus, it is unclear if telomerase directly interacts with TPP1 in human cells.

In this study, we identify separation-of-function mutations in the TEN-domain of hTERT. The mutants retain high catalytic activity, but their interaction with POT1-TPP1 is compromised in vitro. Furthermore, the TEN-domain mutants fail to localize to telomeres and cannot maintain telomere length in vivo. Importantly, a compensatory mutation in the TEL-patch of TPP1 rescues a mutation in the TEN-domain of hTERT both in vitro and in vivo, indicating that the interaction between telomerase and the OB-fold domain of TPP1 is direct. These observations open the door to more directed approaches for targeting the TPP1-telomerase interface as a therapeutic strategy for human diseases.

## Results

### TEN-domain hTERT mutants defective for POT1-TPP1-mediated processivity enhancement

Conserved acidic amino acids on the surface of the TPP1 OB-domain (TEL-patch) are necessary for the recruitment of telomerase to telomeres (*Nandakumar et al., 2012*; *Sexton et al., 2012*; *Zhong et al., 2012*). We hypothesized that a corresponding region of basic amino acids would exist on the surface of hTERT, which directly associates with the TEL-patch via charge–charge and other interactions to recruit telomerase to telomeres (*Figure 1A*).

To identify candidate residues on the TEN-domain of hTERT that might interact with the TPP1 TEL-patch, we performed a multiple sequence alignment of TERT from higher eukaryotes (*Figure 1B*). The alignment revealed a number of conserved basic amino acids; we tested these, as well as some non-conserved basic amino acids in close proximity in the primary sequence. We generated a panel of hTERT mutants, replacing individual or combinations of basic amino acids on hTERT with the acidic amino acid aspartate. Mutant hTERTs and hTR were over-expressed in HEK293T cells and the telomerases were immuno-purified (*Figure 1C*). The hTERT variants were expressed at levels similar to those of wild-type telomerase (*Figure 1D*).

To identify separation-of-function hTERT mutants that do not interact with POT1-TPP1 but retain near wild-type telomerase activity in vitro, we carried out direct telomerase extension assays. (In these assays, dATP, dTTP, and $^{32}$P-dGTP are added to the 3′ end of a DNA oligonucleotide by immuno-purified telomerase and products are separated on a polyacrylamide sequencing gel and visualized by autoradiography. The enzymatic activity of each telomerase is determined by quantifying the total amount of $^{32}$P-dGTP incorporated into reaction products, and RAP is measured by analyzing the distribution of product lengths.) To measure the physical interaction between TPP1 and telomerase, we carried out direct telomerase assays with and without a previously described minimal POT1-TPP1 complex bound to substrate oligonucleotides (*Wang et al., 2007*) and calculated 'RAP stimulation by PT' (equation given in 'Materials and methods'). RAP stimulation by PT depends on the association of telomerase with POT1-TPP1 and therefore provides a biochemical readout for this interaction.

In the hTERT mutant screen, some telomerases had defects in either activity (R72E, R143E, R142E;R143E), RAP (R87E;R91E;K94E, R120E) or had no effect (R142E) (*Figure 1—figure supplement 1*, *Table 1*). However, hTERT K78E retained wild-type activity and processivity (92% and 100% respectively), but RAP stimulation by POT1-TPP1 was reduced to 68% that of wild-type telomerase (*Figure 1E–H*; *Figure 1—figure supplement 2*). Additionally, the R132D mutant was reported to have wild-type activity but failed to localize to telomeres in vivo (*Stern et al., 2012*). We generated a R132E mutant hTERT, which retained moderate activity (73%) but was defective in RAP stimulation by POT1-TPP1 (24% of wild-type hTERT, *Figure 1E–H*; *Figure 1—figure supplement 2*). Furthermore, the K78E;R132E double mutant retained 63% activity but further reduced RAP stimulation by POT1-TPP1 to 18% of wild-type hTERT (*Figure 1E–H*). Two different methods of quantitating RAP gave comparable results (*Figure 1—figure supplement 2*).

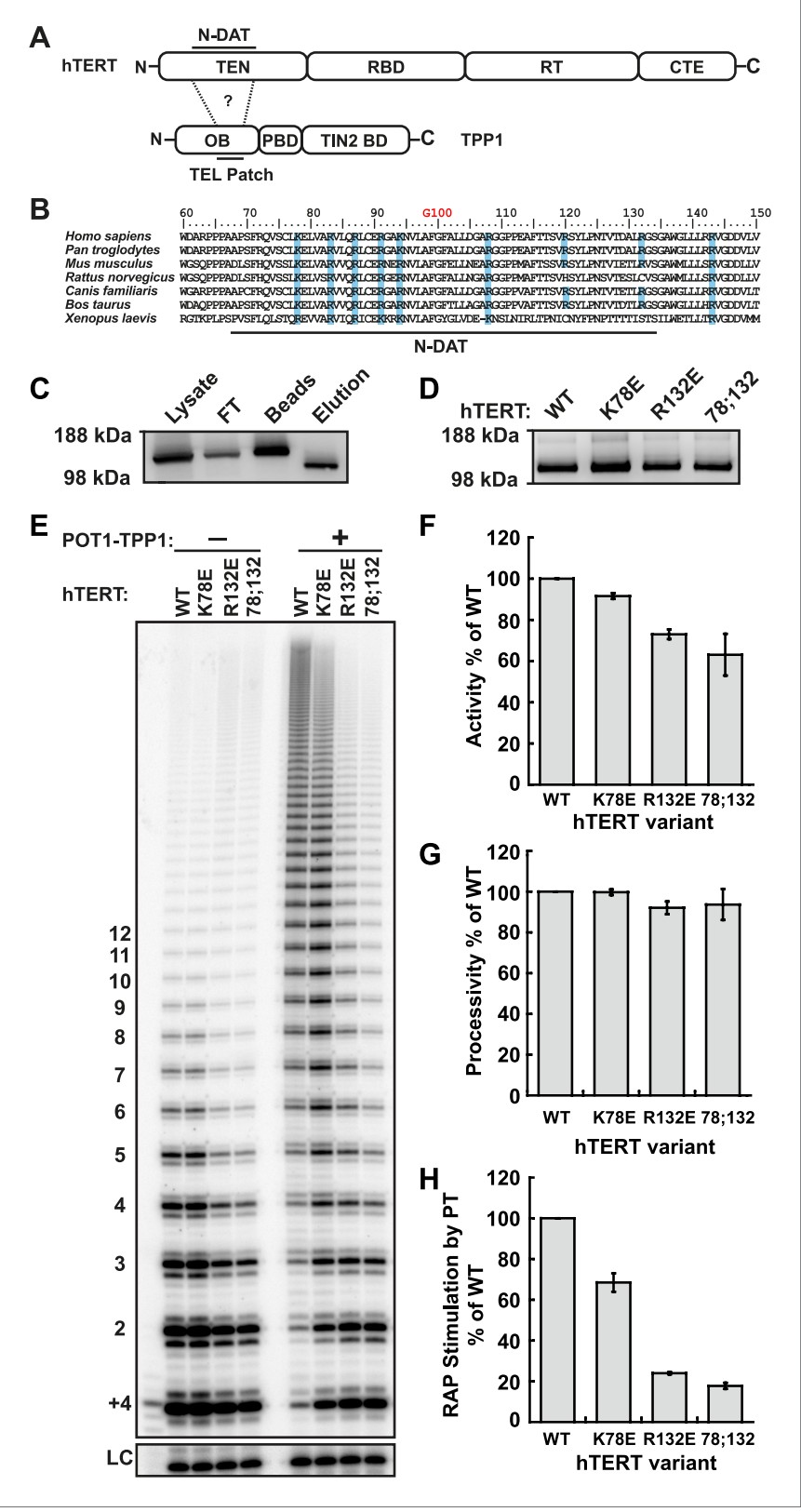

**Figure 1**. Identification of mutants in the TEN-domain of hTERT that affect the interaction with POT1-TPP1 in vitro but not enzymatic activity. (**A**) Domain structures of hTERT and TPP1 proteins. Question mark, proposed interaction between the TEL-patch on TPP1 and the N-DAT region of the TEN-domain of hTERT. In TPP1, the OB-domain (OB)
*Figure 1. Continued on next page*

*Figure 1. Continued*

is followed by the POT1-binding domain (PBD), and finally the TIN2-binding domain (TIN2 BD). For hTERT, the Telomerase Essential N-terminal (TEN) domain is adjacent to the RNA binding domain (RBD), which precedes the reverse transcriptase domain (RT) and finally the C-terminal extension (CTE). (**B**) An alignment of the N-DAT region of the TEN-domain of TERT proteins from selected species. Basic residues conserved in greater than five of the seven species are highlighted blue. The conserved residue G100 is highlighted in orange above the alignment. (**C**) Western blot, probed for hTERT, showing the immuno-purification of telomerase over-expressed in HEK 293T cells. To monitor relative quantities of hTERT, equal fractions of lysate, flow through (FT), IgG bead capture (capture), and cleaved eluate (elution), were analyzed by SDS-PAGE. (**D**) Western blot of the relative quantities of wild-type and mutant hTERTs after immuno-purification of the telomerases. (**E**) Direct telomerase activity assays in the absence and presence of the POT1-TPP1 heterodimer (PT) for wild-type (WT) and mutant telomerases. LC, loading control. +4, oligonucleotide marker corresponding to the addition of the first four nucleotides to primer. Numbers on left, telomeric repeats added. (**F–H**) Bar graphs representing the quantification of activity, RAP, and RAP stimulation (decay method) by wild-type POT1-TPP1. Values are normalized to WT telomerase, and to WT telomerase with WT POT1-TPP1 for RAP stimulation (n = 3, Mean ± SD).

The following figure supplements are available for figure 1:

**Figure supplement 1**. Additional TEN-domain mutants tested in this study.

**Figure supplement 2**. Comparison of methods for quantifying RAP stimulation by POT1-TPP1.

---

An alignment of the human TEN-domain with the *Tetrahymena thermophila* TEN-domain in combination with a secondary structure prediction shows that K78 and R132 are in close proximity to each other, on a surface distinct from the proposed DNA-binding region of the TEN-domain (*Figure 2*, *Figure 2—figure supplement 1*).

These results demonstrate that K78 and R132 within the TEN-domain of hTERT make a much larger contribution to RAP stimulation by POT1-TPP1 than to enzymatic activity in vitro. RAP stimulation defects in K78E and R132E mutants are most likely due to the failure of telomerase to interact with TPP1. Thus, this surface of hTERT is a candidate TPP1-interacting element.

---

**Table 1.** Telomerase TEN domain mutant activity, processivity, and RAP stimulation by wild-type TPP1

| TEN domain mutant | Activity % of WT* | Processivity % of WT | RAP stimulation by PT† % of WT |
|---|---|---|---|
| R72E | 48 ± 3 | 99 ± 1 | 91 ± 2 |
| K78A | 19 ± 2 | 101 ± 7 | 88 ± 2 |
| K78E | 92 ± 1 | 100 ± 1 | 68 ± 5 |
| R87E;R91E;K94E | 74 ± 6 | 75 ± 0.3 | 61 ± 4 |
| R120E | 99 | 96 | 55 |
| K78E;R120E | 82 ± 18 | 87 ± 3 | 38 ± 2 |
| R132E | 73 ± 2 | 92 ± 3 | 24 ± 2 |
| R132E;K78E | 63 ± 10 | 94 ± 8 | 18 ± 1 |
| R142E | 97 ± 7 | 111 ± 10 | 103 ± 4 |
| R143E | 16 ± 12 | 81 ± 3 | N.D.‡ |
| R142E;R143E | 4 ± 1 | N.D.‡ | N.D.‡ |
| V144M | 48 ± 6 | 95 ± 5 | 44 ± 7 |

*Percentage of wild-type telomerase activity, processivity, or RAP stimulation. Activity values normalized to hTERT levels, loading control, ± standard deviation for 2 or more replicates.
†Repeat addition processivity (RAP) stimulation upon addition of WT POT1-TPP1, values relative to WT telomerase with WT POT1-TPP1 (i.e. RAP stimulation by PT % of WT = ((WT PT RAP stimulation of telomerase mutant)/(WT PT RAP stimulation WT telomerase))*100).
‡Not determined (N.D.) due to low telomerase activity.

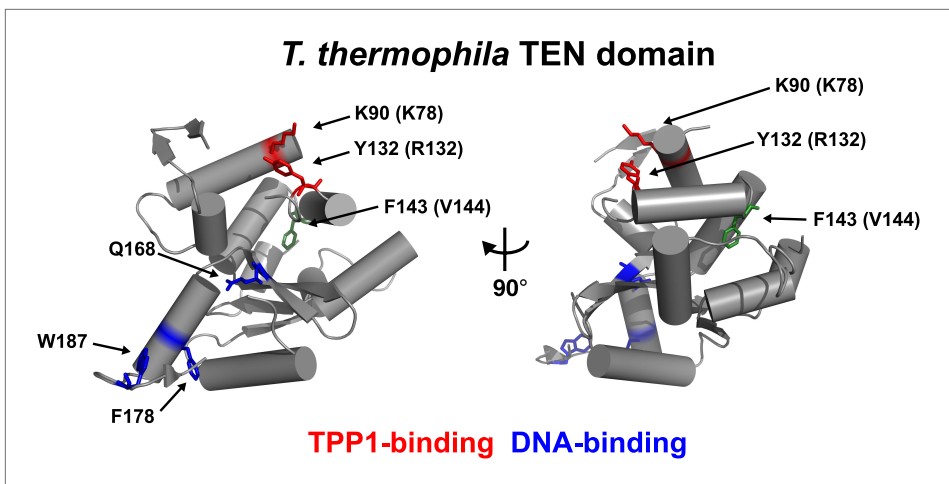

**Figure 2**. Conserved basic residues K78 and R132 are found in close proximity on the surface of the TEN-domain. Cartoon representation of the TEN-domain crystal structure from *Tetrahymena thermophila* (PDB: 2B2A). Amino acid numbers are from *Tetrahymena* (human counterparts in parentheses). Anchor site (DNA binding) residues are highlighted in blue. The positions of the residues corresponding to human K78 and R132 are highlighted in red, and F143 corresponding to human V144 is highlighted in green.

The following figure supplements are available for figure 2:

**Figure supplement 1**. Alignment and secondary structure prediction for human TEN-domain.

## TEN-domain mutants fail to localize to telomeres in vivo

If mutations of K78 and R132 in hTERT affect the interaction of telomerase with TPP1, they should disrupt telomerase localization to telomeres in vivo. To test whether the mutant telomerases are recruited to telomeres, we transiently over-expressed mCherry-tagged hTERT variants and hTR in HeLa cells and determined the subcellular localization of telomerase by immuno-fluorescence (IF). Under all conditions hTERT and hTR were expressed at similar levels, indicating that mutations in the TEN-domain did not affect hTERT or hTR stability (*Figure 3A–B*). As previously described (*Zhong et al., 2012*), wild-type telomerase localized to telomeres and promoted the formation of neo-Cajal bodies at most telomeres, as shown by the co-localization of TRF2, hTERT, and coilin foci (*Figure 3C*). In contrast, all three hTERT mutants tested (K78E, R132E, K78E;R132E) localized to bona fide Cajal bodies, forming ~1–3 large foci per cell that co-localized with coilin but did not localize to telomeres marked by TRF2 (*Figure 3C*). These results demonstrate that TEN-domain mutants that do not interact with POT1-TPP1 in vitro also fail to localize to telomeres in vivo. Furthermore, because Cajal body localization of hTERT requires its association with hTR, our observations indicate that telomerase assembly is unaffected by mutations in the TEN-domain.

## TEN-domain mutants are defective in telomere maintenance in vivo

To determine the effects of the mutant telomerases on telomere maintenance, we generated cell lines stably expressing mCherry-tagged hTERT variants by retroviral transduction. Overexpression of hTERT alone leads to a moderate increase in telomerase activity per cell, more closely resembling endogenous telomerase levels than those obtained by overexpression of both hTERT and hTR (*Cristofari et al., 2007*; *Xi and Cech, 2014*). Thus, while the absolute expression level of mCherry-hTERT varied between the stable cell lines (*Figure 4A*), telomerase immuno-purified from the cell lines had comparable activities per cell, and these activity levels were threefold to fourfold higher than levels in untransfected control cells (*Figure 4B*). This result confirmed that mutations in the TEN-domain did not strongly reduce telomerase activity. The exogenous hTERT was overexpressed relative to endogenous hTERT, which was not detectable by western blot (*Figure 4A*). Therefore, the majority of endogenous hTR should be assembled into telomerase RNPs containing the mutant hTERT protein (*Figure 4A*).

To analyze the effect of mutant hTERT expression on telomere length, DNA isolated from stable cell lines over the course of 8 weeks after viral transduction was subjected to telomeric restriction

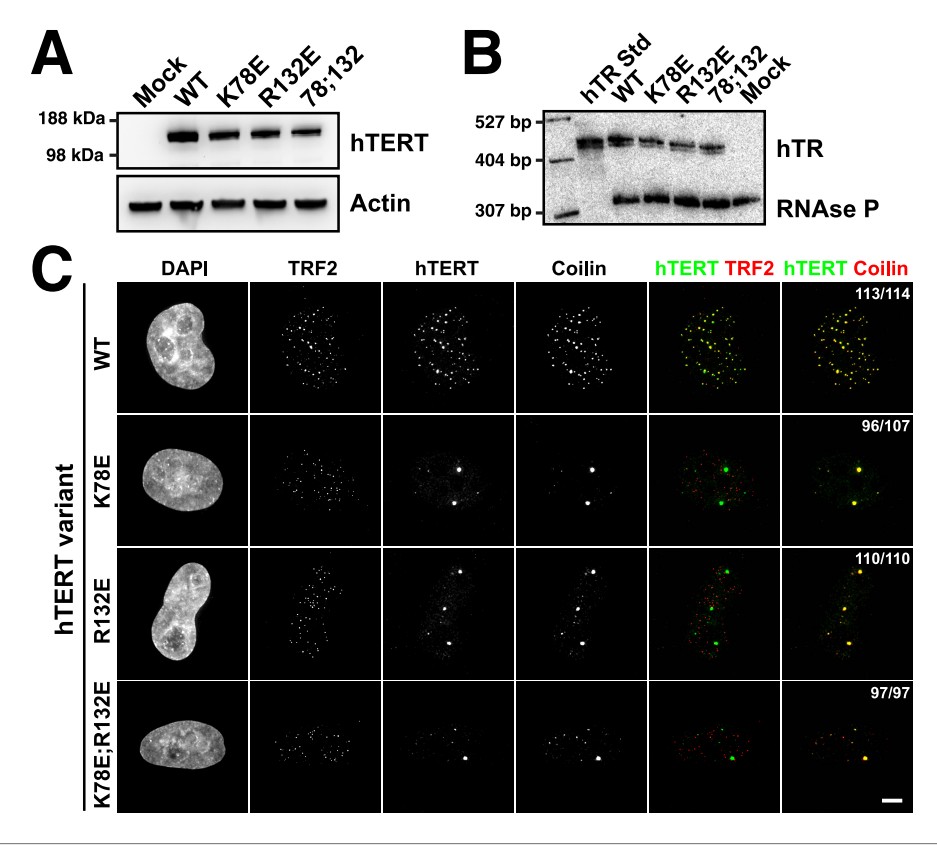

**Figure 3**. TEN-domain mutations disrupt telomere localization of telomerase. (**A**) Western blots of lysates of HeLa cells transfected with expression plasmids for various hTERT alleles and hTR, probed with an antibody against hTERT. Actin was used as a loading control. (**B**) Northern blots of RNA isolated from HeLa cells transfected with expression plasmids for various hTERT alleles and hTR, using probes for hTR. In vitro transcribed hTR (500 pg) was used as positive control. Blots were probed for RNase P RNA as loading control. (**C**) Immuno-fluorescence (IF) analysis of HeLa cells transiently transfected with mCherry-hTERT and hTR plasmids. Cells were fixed and probed with antibodies against mCherry, coilin, and TRF2 to visualize telomerase, Cajal bodies and telomeres, respectively. Images were deconvolved. Numbers indicate the fraction of cells analyzed showing the displayed phenotype (scale bar = 5 µm).

fragment (TRF) analysis by Southern blot. The telomere length of the parental HeLa cell line was stable over the course of the experiment at ~6.6 kb (*Figure 4C*). Expression of wild-type hTERT caused a progressive increase in telomere length to ~14.8 kb (*Figure 4C*). In contrast, TEN domain mutants K78E and R132E failed to elongate telomeres, despite the fact that telomerase enzymatic activity per cell was higher than in the parental HeLa cell line (*Figure 4B,C*). In addition the TEN domain mutant K78E;R132E also failed to elongate telomeres, but due to its reduced activity (*Figure 1F*), we cannot conclude that this is exclusively due to a localization defect. These observations reinforce the conclusion that telomerase with K78E or R132E mutations in the TEN-domain fails to elongate telomeres due to an inability to localize to telomeres, not an activity defect.

## A compensatory mutation in TPP1 rescues the mutant hTERT—TPP1 interaction in vitro

Taken together, our experiments demonstrate that mutations in the TEN-domain of hTERT recapitulate the effects of mutations in the TEL-patch of TPP1 (i.e., loss of telomerase recruitment to telomeres in vivo, and reduced POT1-TPP1 RAP stimulation in vitro). If the stimulation of telomerase RAP and telomerase recruitment to telomeres are the manifestation of a direct interaction between charged residues in the TEN-domain of hTERT and the OB-fold of TPP1, a charge-swap mutation in the OB-fold of TPP1 could rescue the TEN-domain mutation (*Figure 5A*). On the other hand, if a bridging factor

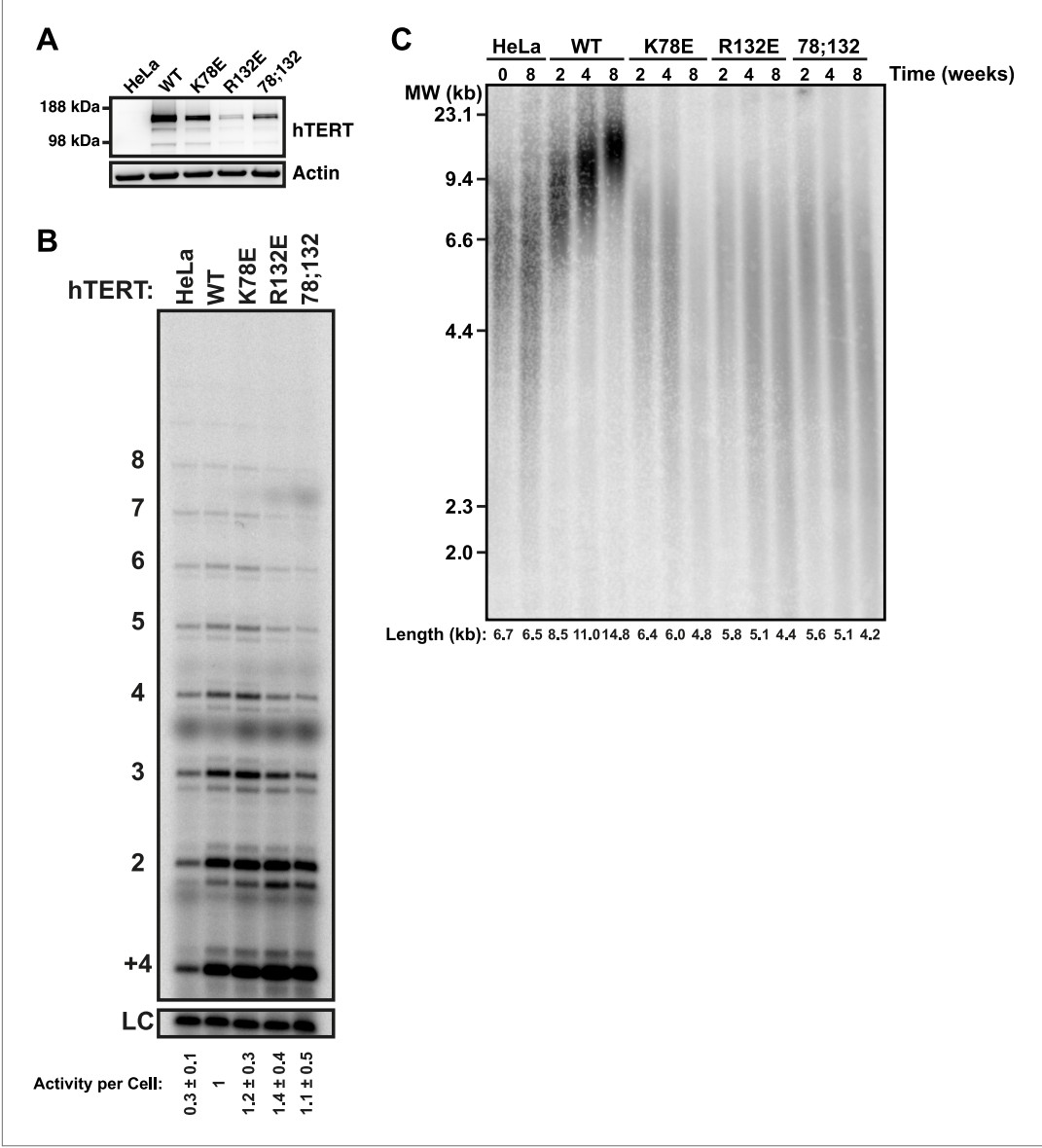

**Figure 4**. TEN-domain mutants that do not localize to telomeres fail to elongate telomeres in vivo. (**A**) Western blot probed for hTERT and for Actin as a loading control, showing hTERT expression in lysates of parental HeLa cells and cell lines stably expressing mCherry-hTERT variants. hTR was not ectopically expressed. (**B**) Direct enzyme assay of telomerase immuno-purified from lysates, generated using equal number of cells, from parental HeLa or cell lines stably expressing mCherry-hTERT variants. Activity per cell relative to WT hTERT overexpressing cells (n = 4, Mean ± SD, p < 0.05). (**C**) Telomeric restriction fragment Southern blot of cell lines stably expressing mCherry-hTERT variants over the time course of 8 weeks.

were necessary for the interaction of TPP1 with hTERT, individually deleterious mutations in both TPP1 and the TEN-domain should have an additive negative impact on the TEN-TPP1 interaction when combined (*Figure 5A*).

To test this hypothesis, we generated a number of mutant TPP1N proteins, changing individual conserved negatively charged residues in the TEL-patch to the basic amino acid lysine. Mutant TPP1 proteins were co-purified with POT1 to apparent homogeneity (*Figure 5B*). All mutant POT1-TPP1 complexes had identical elution profiles from the sizing column (*Figure 5C*). The presence of a single elution peak indicated that all TEL-patch mutant TPP1 proteins were not globally misfolded and formed heterodimers with POT1 (*Figure 5C*).

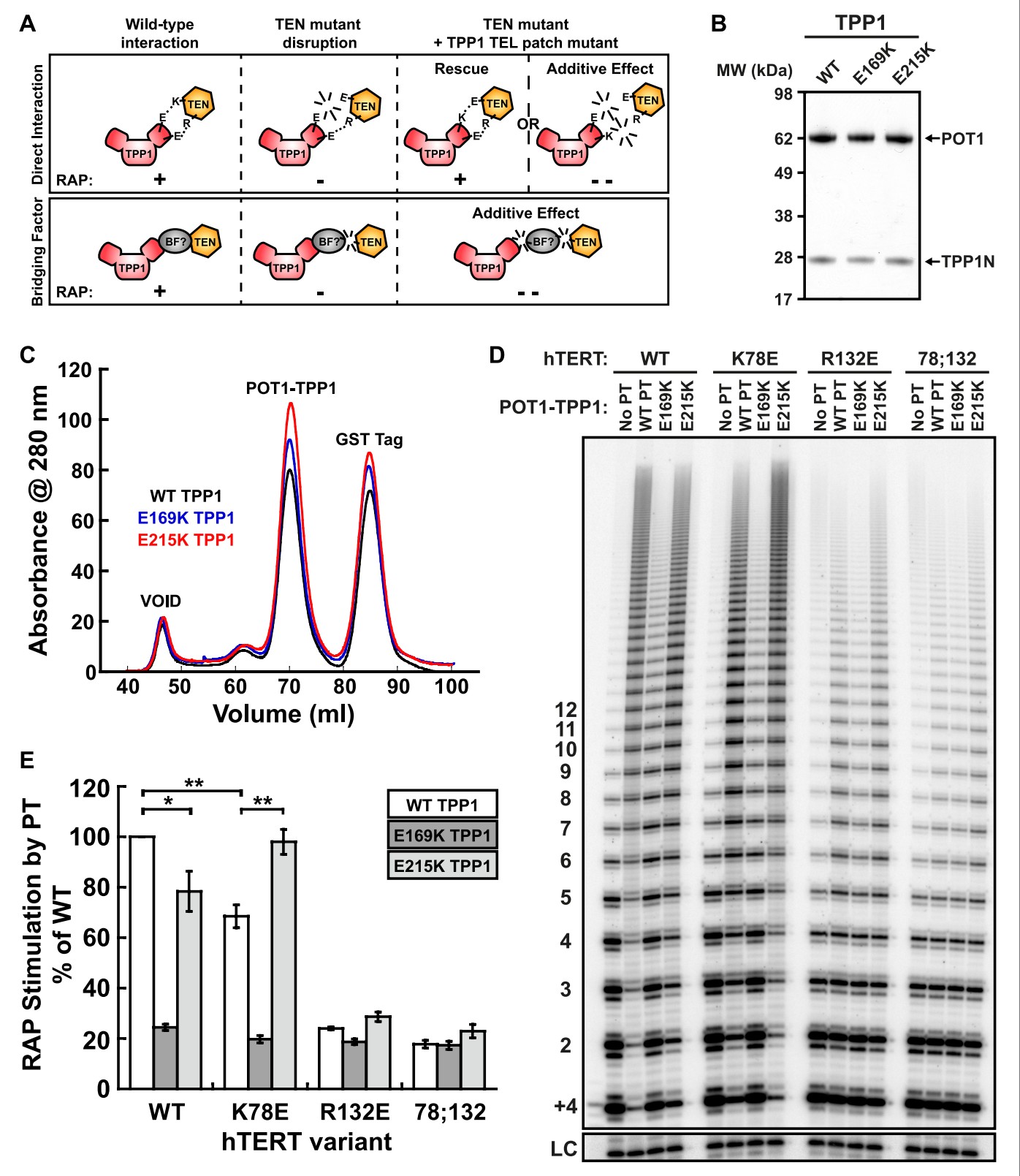

**Figure 5**. A compensatory mutation in the TEL-patch of TPP1 rescues RAP stimulation of TEN-domain mutant telomerase in vitro. (**A**) Schematic of the charge-swap experiment to test the interaction between specific amino acids on the TEL-patch on TPP1 and the N-DAT region of the TEN-domain of
*Figure 5. Continued on next page*

*Figure 5. Continued*

hTERT. Predicted experimental outcomes are illustrated for two competing models: a direct TPP1 telomerase interaction and an interaction bridged by a yet-unidentified factor (BF?). (**B**) Coomassie-stained gel of WT, E169K, and E215K TPP1 co-purified with wild-type POT1. (**C**) Overlays of the Superdex 200 16/60 sizing column chromatograms for wild-type and mutant TPP1-POT1 complexes. (**D**) Direct telomerase activity assay to test the rescue of various telomerase containing WT, K78E, R132E, and K78E;R132E mutant hTERTs in the absence of POT1-TPP1 (No PT) or WT, E169K, and E215K TPP1 in complex with wild-type POT1. LC and +4 marker as in *Figure 1E*. (**E**) Quantification (decay method) of the RAP stimulation by POT1-TPP1 (PT) relative to the stimulation of wild-type telomerase with wild-type POT1-TPP1 (n = 3, Mean ± SD, *p < 0.05, **p < 0.01, Student's *t* test).

The following figure supplements are available for figure 5:

**Figure supplement 1**. The charge-swap is statistically significant by alternative quantitation methods.

**Figure supplement 2**. E215K TPP1 rescues RAP stimulation of K78E mutant telomerase produced in RRLs.

**Figure supplement 3**. The IPF allele V144M is deficient in RAP stimulation by TPP1.

To test the ability of the mutant TPP1 proteins to bind TEN-domain mutants, we used direct telomerase enzyme assays to measure RAP stimulation by POT1-TPP1. Strikingly, TPP1 E215K fully rescued the processivity defect of hTERT K78E; processivity increased from 68% to 98% relative to wild-type telomerase and wild-type TPP1 (*Figure 5D,E*). Importantly, the non-cognate combinations of (i) wild-type telomerase and E215K TPP1 and (ii) K78E telomerase with wild-type TPP1 showed a similar reduction of RAP stimulation, to 78% and 68% of the full level, respectively (*Figure 5D,E*). TPP1 E169K gave dramatic reductions in RAP and did not stimulate any of the tested telomerases above 24% of wild-type (*Figure 5D,E*). hTERT mutant R132E and the double mutant K78E;R132E were stimulated at levels less than 23% of wild-type by all TPP1 proteins tested (*Figure 5D,E*). An alternative method for quantification of RAP stimulation by POT1-TPP1 gave similar results (*Figure 5—figure supplement 1*). Furthermore, to rule out the possibility of a bridging factor that co-purifies with telomerase from HEK293 T cells, we carried out the charge-swap experiment with telomerase purified from rabbit reticulocyte lysates (RRLs). The charge-swap rescue was statistically significant when using telomerase from RRLs, although the absolute PT stimulation of RRL telomerase was diminished (*Figure 5—figure supplement 2*).

To further illustrate the specificity of the charge-rescue, a disease allele V144M found in some cases of idiopathic pulmonary fibrosis (IPF) (*Tsakiri et al., 2007*) was tested for interaction with POT1-TPP1. V144M telomerase retained moderate telomerase activity and high processivity (48% and 95% of wild-type telomerase, respectively) but POT1-TPP1 RAP stimulation was significantly decreased to 61% when compared to wild-type telomerase (*Table 1*, *Figure 5—figure supplement 3*). Furthermore, TEL patch mutants E169K and E215K each resulted in an additive loss of RAP stimulation when tested with disease mutant V144M, to 28% and 37% respectively (*Figure 5—figure supplement 3*). Thus, the specific and compensatory nature of the hTERT K78E and TPP1 E215K mutant combination strongly suggests that K78 on the TEN-domain of hTERT directly interacts with E215 on the TEL-patch of TPP1 to stimulate RAP in vitro.

## A compensatory mutation in the TEL-patch of TPP1 restores the telomerase TPP1-OB interaction at a non-telomeric locus in vivo

To analyze mutant hTERT-TPP1-OB-fold interactions in vivo, we utilized a Lac-repressor (LacI) assay previously described (*Zhong et al., 2012*). The TPP1-OB-fold domain was inserted between GFP and the LacI protein (*Figure 6A*). GFP-TPP1-OB-LacI was expressed alongside mCherry-tagged hTERT and hTR in the U2OS 2-6-3 cell line, which contains a single Lac-operator DNA array on chromosome 1 (*Janicki et al., 2004*), thereby tethering the OB-fold domain of TPP1 to a non-telomeric chromosomal locus (*Figure 6A*). The interaction between telomerase and the TPP1-OB-fold domain was assessed by co-localization of GFP- and mCherry foci in cell nuclei. Wild-type telomerase co-localized with the TPP1-OB-fold domain in ~90% of nuclei. In contrast, in cells expressing K78E, R132E, or K78E;R132E telomerase, the fraction of TPP1-OB-fold domain foci showing telomerase signal was reduced to ~55%, ~30%, and 0%, respectively (*Figure 6B,C*, additional examples in *Figure 6—figure supplement 1*). The reduction in co-localization demonstrated that mutation of the hTERT TEN-domain interferes with the interaction between telomerase and the TPP1-OB-fold domain in vivo.

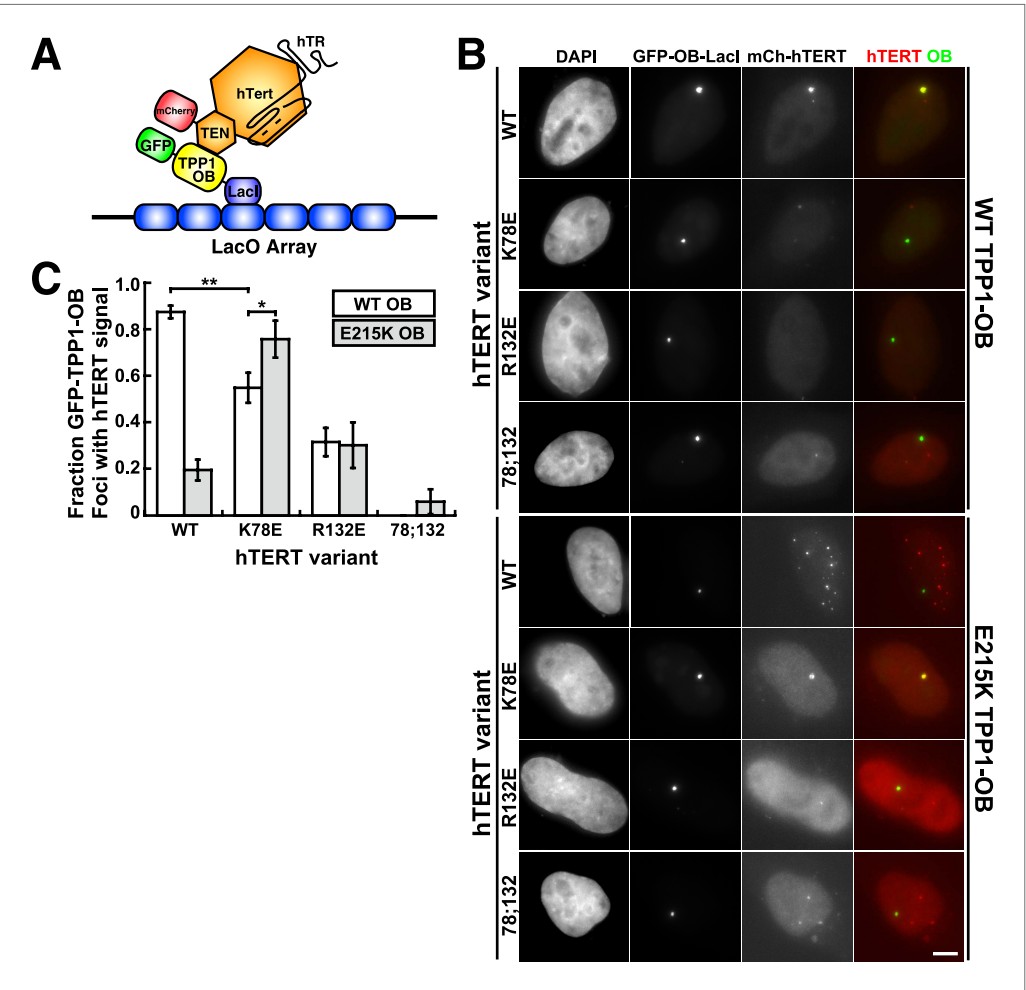

**Figure 6**. A compensatory mutation in the TEL-patch of TPP1 rescues TEN-domain mutant telomerase binding to the TPP1 OB-domain in vivo. (**A**) Model showing the experimental design. Fusion of the OB-domain of TPP1 to the lac repressor (LacI) recruits the OB-domain to a single non-telomeric chromosomal locus (LacO array), allowing the interaction between telomerase and the TPP1 OB-domain to be assessed by co-localization of GFP (TPP1 OB) and mCherry (hTERT) in cell nuclei. (**B**) Fluorescence images showing the localization of GFP-TPP1-OB-LacI and mCherry-hTERT fusion proteins in cell nuclei stained by DAPI. Cells were fixed, permeabilized, and stained with DAPI. The intrinsic GFP- and mCherry-fluorescence was used to detect GFP-TPP1-OB-LacI and mCherry-hTERT (scale bar = 5 μm). (**C**) Quantification of the experiments shown in (**B**), showing the fraction of nuclei with co-localization of GFP- and mCherry-foci (n = 3, 117–176 nuclei total, Mean ± SD, *p < 0.05, **p < 0.01, Student's *t* test).

The following figure supplement is available for figure 6:

**Figure supplement 1**. Additional examples of U2OS 2-6-3 cells expressing GFP-TPP1 OB-LacI and mCherry-telomerase.

To determine whether mutations introduced in the OB-fold domain of TPP1 can compensate for the mutations introduced in hTERT in vivo, we carried out the LacI-assay with E215K TPP1-OB. As shown in *Figure 6B,C*, the co-localization of wild-type telomerase with the E215K TPP1-OB-fold domain was greatly reduced; instead, wild-type telomerase localized to many foci distinct from the LacI site, presumably telomeres. In contrast, introduction of the E215K mutation significantly increased the co-localization of K78E telomerase with the TPP1-OB-fold domain (p < 0.05). Thus the E215K mutation in the TPP1-OB-fold compensates for the presence of the K78E mutation in hTERT in vivo. Importantly, E215K TPP1 had no effect on the localization of R132E and K78E;R132E telomerases, demonstrating that E215K specifically compensates for the presence of the K78E mutation in hTERT

(*Figure 6B,C*). If another protein were bridging telomerase and TPP1, any combination of two defective mutations should be even more defective. Thus, finding a compensatory double mutant demonstrates that telomerase is recruited to telomeres by a direct interaction between the TEN-domain of telomerase and the TPP1-OB-fold domain in part by a contact formed between K78 in the TEN-domain and E215 in the TPP1-OB-fold domain.

## TPP1 E215K rescues telomere maintenance in cells expressing K78E hTERT

To address whether TPP1 E215K could rescue recruitment of hTERT K78E to telomeres, we transduced cell lines stably expressing WT and K78E mCherry-hTERT with retrovirus expressing wild-type (WT), E169K, and E215K TPP1-FLAG alleles. The resulting cell lines expressed both TPP1-FLAG and mCherry-hTERT proteins (*Figure 7A–B*). Importantly, retroviral transduction was carried out after substantial telomere erosion had occurred due to the expression of K78E hTERT (*Figure 4C*, transduction carried out after 8 weeks).

To test whether the expression of E215K TPP1 rescued telomere maintenance in cells expressing K78E hTERT, we carried out TRF analysis by Southern blot (*Figure 7C*). Telomere length in the cell line expressing only K78E hTERT was constant at ~3.4 kb during the 6-week experiment. Expression of WT TPP1 led to an increase of telomere length to ~6.2 kb. Importantly, expression of E215K TPP1 triggered a more pronounced increase in telomere length to ~7.7 kb over the same time period, while E169K gave lead to a subtle increase to ~4.9 kb. These observations demonstrate that expression of E215K TPP1 in cells expressing K78E hTERT rescues telomere length maintenance, most likely by driving telomerase recruitment to telomeres.

## Discussion

Telomerase recruitment to telomeres is a crucial step in telomere maintenance. Here, we demonstrate that K78 and R132 within the TEN-domain of hTERT are critical residues that mediate the direct interaction of telomerase with the TEL-patch of TPP1, recruiting telomerase to telomeres (*Figure 8*). Mutation of either interaction partner results in sequestration of telomerase in Cajal bodies and failure to elongate telomeres in vivo. Our observations provide a molecular mechanism explaining the failure of previously described N-DAT mutants to function in vivo. Additionally, the identification of the interface between telomerase and the telomere will allow more targeted approaches to alter telomerase recruitment as a potential therapeutic approach for human diseases.

### Specific hTERT TEN-domain amino acids are necessary for telomerase recruitment to telomeres

During the S-phase of the cell cycle, telomerase moves from Cajal bodies to telomeres to counteract the progressive shortening of the telomere that occurs during DNA replication (*Jady et al., 2006*; *Tomlinson et al., 2006*; *Cristofari et al., 2007*; *Stern et al., 2012*). The regulatory mechanisms underlying this process are not well understood, in part because the necessary molecular interfaces are not fully defined. On the telomere side, TPP1 is well established as being necessary for the recruitment of telomerase (*Abreu et al., 2010*; *Nandakumar et al., 2012*; *Sexton et al., 2012*; *Zhong et al., 2012*). Localization of telomerase to an artificial non-telomeric focus by the TPP1 OB-domain (*Zhong et al., 2012*) suggests that the TPP1 OB-domain is the minimal sufficient telomerase recruitment module of the shelterin complex. The TEL-patch provides a molecular surface that mediates telomerase binding, as single mutations in the TEL-patch result in loss of telomere recruitment and of RAP stimulation of telomerase by TPP1 (*Nandakumar et al., 2012*; *Sexton et al., 2012*; *Zhong et al., 2012*).

While the telomeric half of the interface required for telomerase recruitment to telomeres is well understood, the molecular determinants on telomerase required for telomere localization were thus far ill defined. Several lines of evidence, including naturally occurring disease-associated alleles of hTERT (V144M) (*Tsakiri et al., 2007*) and synthetic mutations (G100V, R132D, N-DAT), have implicated the TEN-domain of hTERT in telomerase recruitment to telomeres (*Armbruster et al., 2001*; *Zaug et al., 2010*; *Stern et al., 2012*). Unfortunately, due to the involvement of the TEN-domain in catalysis and RAP, many of these mutations also alter the enzymatic activity of telomerase. Thus, TEN-domain mutants can have pleotropic effects and may not specifically disrupt the telomere–telomerase interface (*Zaug et al., 2010*, *2013*). Our results demonstrate that a region in the TEN-domain including K78, R132, and V144 is required for telomerase recruitment in vivo and POT1-TPP-mediated RAP

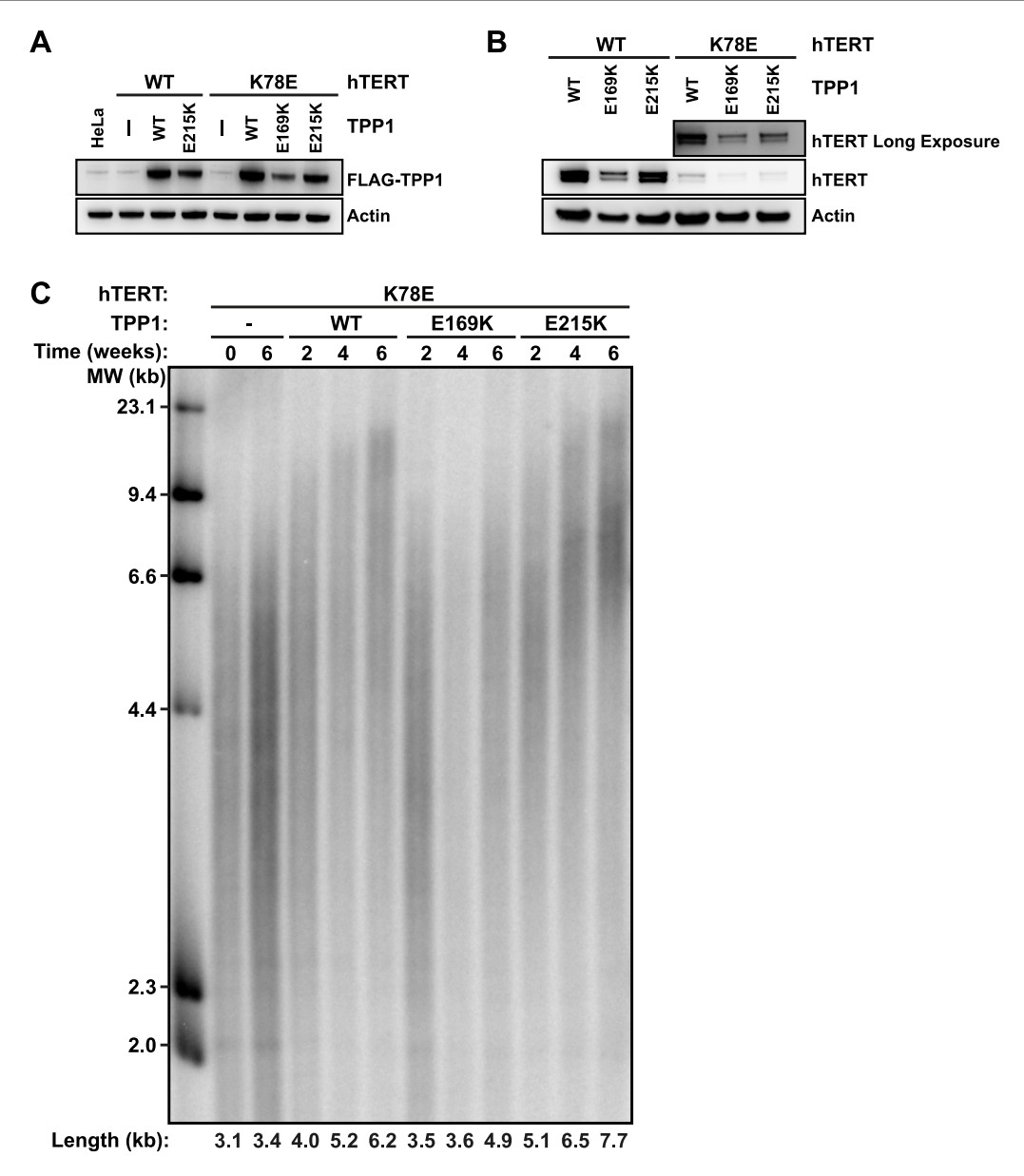

**Figure 7**. TPP1 E215K rescues telomere maintenance in cells expressing hTERT K78E. (**A**) Western blot probed for TPP1-FLAG and for Actin as a loading control, showing TPP1 expression in lysates of parental HeLa cells and cell lines stably expressing TPP1-FLAG and mCherry-hTERT variants. (**B**) Western blot probed for hTERT and for Actin as a loading control, showing hTERT expression in lysates of cell lines stably expressing TPP1-FLAG and mCherry-hTERT variants. (**C**) Telomeric restriction fragment Southern blot of cell lines stably expressing K78E mCherry-hTERT and TPP1-FLAG variants over the time course of 6 weeks.

stimulation in vitro, albeit to varying degrees. The differences in RAP stimulation may reflect the importance of the respective residue in an interaction with TPP1. The recruitment defects and failure to maintain telomeres in vivo occur in the context of wild-type levels of activity, suggesting that the mutants described in this study are separation-of-function alleles that dissociate enzymatic activity from recruitment. Importantly, the recruitment defects of K78E and R132E telomerases recapitulate those of TEL-patch mutant TPP1 proteins (*Zhong et al., 2012*; *Nandakumar et al., 2012*). The interaction between the TEL-patch of TPP1 and the TEN-domain of hTERT provides a molecular mechanism to explain the failure of N-DAT mutations to immortalize cells.

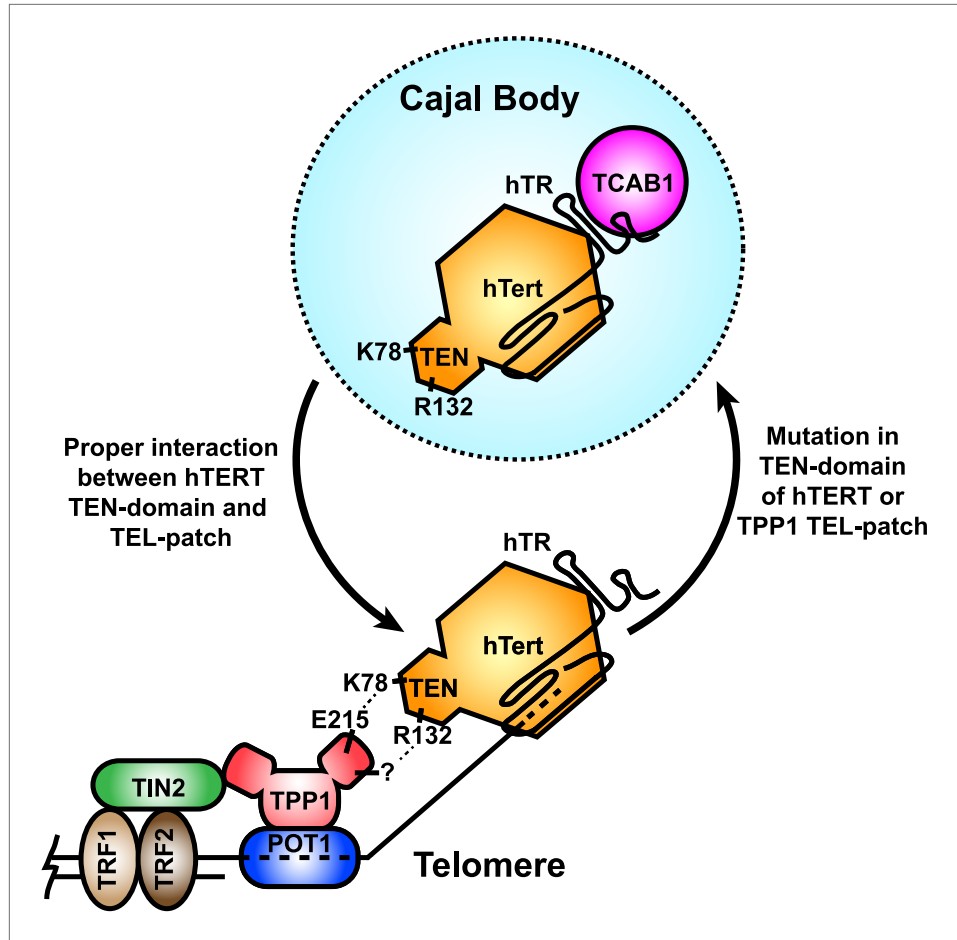

**Figure 8**. A model for the recruitment of telomerase to telomeres through the direct interaction between the TEL-patch of TPP1 and the hTERT TEN-domain. Throughout the cell cycle, telomerase associates with TCAB1 and localizes to Cajal bodies. During late S/G2, telomerase is recruited to telomeres through a direct interaction between K78 on the TEN-domain of hTERT and E215 within the TEL-patch of TPP1. R132 in the TEN-domain also participates in the interaction, but the corresponding residue on the TEL-patch and the nature of the interaction remains unknown (?). TPP1 is recruited to telomeres through interaction with TIN2 and further stabilized by the single-stranded DNA-binding protein POT1. TRF1 and TRF2 comprise the double-stranded DNA binding proteins of shelterin. Mutations in the OB-domain of TPP1 or TEN-domain of hTERT that disrupt the interaction are sufficient to prevent recruitment to telomeres, instead resulting in the sequestration of telomerase in Cajal bodies.

## Telomerase recruitment to telomeres in human and other organisms

Our observation that charge-swap mutations on the TEN-domain of hTERT and the TEL-patch of TPP1 restore telomerase recruitment indicates that these elements interact directly; for other examples see *Jucovic and Hartley (1996)*; *Tansey and Herr (1997)*; *Pennock et al. (2001)*. If instead there were an 'adapter' protein bridging the two elements, the combination of two individually deleterious mutations would lead to an additive loss of function (*Figure 5A*). We note that hTERT K78E and TPP1 E215K lead to similar reductions in RAP stimulation when combined with wild-type TPP1 and wild-type hTERT, respectively. This observation adds additional support to the model that hTERT K78E and TPP1 E215K eliminate reciprocal residues of the same molecular interaction. In contrast, the V144M mutation in hTERT and the E215K mutation in the TEL-patch of TPP1 display an additive reduction in RAP stimulation, indicating that they affect separate components of the telomerase–TPP1 interface.

In other organisms, however, the association of telomerase with the telomere occurs through different mechanisms. For example, a bridging factor Ccq1 appears to be necessary for recruitment of telomerase to the telomeric protein Tpz1 (TPP1 homolog) in *S. pombe* (*Miyoshi et al., 2008*; *Moser et al., 2011*). A homologue of Ccq1 has not been identified in humans (*Nandakumar and Cech,*

*2013*). Another variation occurs in *Saccharomyces cerevisiae*, where Est3 shares structural conservation with TPP1 (*Yu et al., 2008*; *Rao et al., 2014*). Est3 is part of the telomerase holoenzyme instead of the telomeric cap (*Hughes et al., 2000*). Est3 is thought to make direct contacts with the TEN-domain of Est2, the yeast TERT (*Friedman et al., 2003*; *Talley et al., 2011*; *Yen et al., 2011*). However, Est3 appears to stimulate telomerase activity rather than act as a recruitment factor (*Talley et al., 2011*). Recruitment is instead mediated by the interaction of the Est1 subunit of the telomerase holoenzyme and Cdc13 at the telomere (*Evans and Lundblad, 1999*), and charge-swap mutations indicate that this interaction is direct (*Pennock et al., 2001*). Thus, in budding yeast the interaction between a TPP1-like OB-domain protein (Est3) and the TEN-domain of TERT (Est2) is conserved, but the recruitment mechanism differs significantly from that in humans.

In addition to the interaction between TPP1 and hTERT in humans, other factors contribute to the productive recruitment of telomerase to the telomere. Depletion of TIN2 from shelterin (*Abreu et al., 2010*) and perturbation of the TCAB1-hTR interaction (*Stern et al., 2012*) both result in a reduction of telomerase association with the telomere. In spite of the aforementioned defects, these proteins likely play important but indirect roles in telomeric recruitment. TCAB1 is necessary for telomerase maturation and trafficking to Cajal bodies (*Venteicher et al., 2009*), while TIN2 is required for TPP1 localization to telomeres (*Abreu et al., 2010*).

Previous work also suggested that the C-DAT region of the CTE of hTERT contributed to telomerase recruitment (*Banik et al., 2002*; *Zhong et al., 2012*). However, synthetic C-DAT mutants have severe activity defects (*Huard et al., 2003*; *Jurczyluk et al., 2011*). In addition, disease-associated mutations in and near C-DAT have varying effects on activity and cellular localization of telomerase. E1117X has ~10% of wild-type activity and appears to be sequestered to Cajal bodies in vivo (*Tsakiri et al., 2007*; *Zhong et al., 2012*); in contrast, F1127L has reasonable activity (~70%) and localizes to telomeres, and it is stimulated by POT1-TPP1 (*Zhong et al., 2012*; *Zaug et al., 2013*). Thus, CTE mutants do not have clear recruitment defects dissociated from activity loss, despite their importance based on disease association. In contrast, the observation that a mutant TPP1-OB-domain increases the localization of a mutant hTERT to a non-telomeric locus in vivo, while it strongly decreases the association with wild-type hTERT, demonstrates that a direct TEN-domain-TEL-patch interaction is necessary and may be sufficient for telomerase recruitment to telomeres in humans.

### The interface of telomerase recruitment as a therapeutic target

The identification of a direct protein–protein interaction surface between telomerase and the telomere is key to identifying modulators of telomerase recruitment as potential therapeutic agents. Multiple disease mutations associated with IPF—P33S, L55Q, Pro112ProfsX16, and V144M—cluster in TEN-domain of hTERT. Mutation carriers have significantly shorter telomeres than non-carrier relatives (*Armanios et al., 2007*; *Tsakiri et al., 2007*). These mutations have variable impacts on telomerase enzymatic activity (*Armanios et al., 2007*; *Tsakiri et al., 2007*; *Zaug et al., 2013*). Importantly, V144M is defective in TPP1-mediated RAP stimulation in vitro (this study) and fails to localize to telomeres in vivo (*Zhong et al., 2012*). Given the propensity of both synthetic and IPF-associated mutations in the TEN-domain of hTERT to disrupt telomere localization, it is likely that some cases of IPF are directly caused by a decrease in telomerase recruitment to telomeres. Agonists of the TEL-patch-TEN-domain interaction that compensated for these mutations and attenuated telomere shortening associated with the disease might be pharmaceutically important.

In contrast to the loss of telomerase function associated with IPF, reactivation of telomerase is a hallmark of most human cancers. Inhibitors that disrupt telomerase recruitment to telomeres would provide an additional approach to target telomerase activity in cancer cells. In support of this idea, disruption of the TEL-patch diminishes cell growth and triggers apoptosis in HeLa cells, and this effect is exacerbated in combination with a small molecule inhibitor of telomerase activity (*Nakashima et al., 2013*). Our finding that the TEN-domain of hTERT interacts directly with the TEL-patch of TPP1 to bring telomerase to the telomere defines a key interface and provides a direct target for the design of novel therapeutic inhibitors of telomerase action.

## Materials and methods

### Telomerase purification

Telomerase was overexpressed in HEK293T cells and purified as previously described (*Sauerwald et al., 2013*). Briefly, 1 ml whole cell lysates of 50–60 × $10^6$ HEK293T cells overexpressing ZZ-TEV-3xFlag-hTERT

variants and hTR were incubated with ~350 µl IgG-Sepharose at 4°C for 3–4 hr. Following a wash with 50 ml wash buffer (20 mM HEPES-KOH pH 7.9, 300 mM KCl, 2 mM MgCl$_2$, 1 mM EDTA, 1 mM DTT, 1 mM PMSF, 0.1% Triton X-100, 10% glycerol), telomerase was eluted from the beads by cleavage with TEV-protease. The telomerase bound to IgG-Sepharose was incubated with ~500 µl of elution buffer (20 mM HEPES-KOH pH 7.9, 150 mM KCl, 2 mM MgCl$_2$, 1 mM EDTA, 1 mM DTT, 1 mM PMSF, 0.1% Triton X-100, 10% glycerol) supplemented with 5 µl of Act-TEV (Life technologies, Carlsbad, CA) and 5 µl RNAsin+ (Promega, Madison, WI) overnight at 4°C. Telomerase-containing eluate was aliquoted, snap frozen in liquid nitrogen, and stored at −80°C until use.

To purify endogenous telomerase from HeLa cells and HeLa cell lines overexpressing mCherry-hTERT variants, telomerase was immuno-purified using a sheep polyclonal antibody (a kind gift from S Cohen) as previously described (*Cohen and Reddel, 2008*). Briefly, cell lysates of 1 × 10$^6$ cells were incubated with 40 µg of anti-hTERT antibody, which was captured using protein-G agarose. Telomerase was eluted using the peptide antigen used to raise the antibody. Eluates were used for direct telomerase extension assays.

Telomerase was produced in RRLs as previously described (*Zaug et al., 2013*) using ProA-tagged hTERT and purified using the same protocol as telomerase from HEK293T cells described above.

## Western blot

Western blots were carried out using 4–12% polyacrylamide Bis Tris Glycine gels (Life Technolgies) and antibodies against hTERT (Ab32020; 1:1000; Abcam, UK), beta-Actin (A5441; 1:5000; Sigma), and an HRP-conjugated FLAG-antibody (A8592; 1:1000; Sigma). Secondary antibodies (Jackson ImmunoResearch, West Grove, PA) were used at 1:5000. Detection was carried out using SuperSignal Western Pico Chemiluminescence substrate (ThermoFisher Scientific, Waltham, MA).

## POT1-TPP1N purification

Cell pellets from insect cells overexpressing GST-POT1 and *Escherichia coli* cells overexpressing 6xHis-SUMO-TPP1N were lysed by sonication in lysis buffer (PBS supplemented with 250 mM KCl, 1 protease inhibitor tablet [ThermoFisher Scientific], 1 mM PMSF). Lysates were cleared by centrifugation at 40,000×*g* for 35 min at 4°C. Equal volumes of the insect-cell lysates expressing GST-POT1 and bacterial lysates expressing 6xHis-SUMO-TPP1N were combined and incubated with 1 ml of glutathione–sepharose resin for 1 hr at 4°C. After washing three times with 50 ml of GST-wash buffer (PBS supplemented with 250 mM KCl, 1 mM DTT, and 1 mM PMSF), the POT1-TPP1N complex was eluted with 3 ml of GST-elution buffer (50 mM Tris pH 8.1, 75 mM KCl, 10 mM glutathione). The elutions were supplemented with 0.5% w/w PreScission protease and SUMO-protease and incubated on ice for 0.5–1 hr, followed by size-exclusion chromatography on a Superdex 200 16/60 column (GE Healthcare, UK) in gel-filtration buffer (50 mM Tris pH 7.0, 150 mM NaCl, 1 mM DTT). Size-exclusion fractions were pooled, concentrated, supplemented with 10% glycerol vol/vol, snap-frozen in liquid nitrogen and stored at −80°C until use.

## Direct telomerase activity assay

Direct telomerase activity assays were carried out as previously described (*Zaug et al., 2013*), with slight modifications. The final 20 µl reactions contained 2 µl eluted telomerase extract, 500 nM primer, 500 nM POT1-TPP1 complex, and 150 mM KCl. Direct telomerase assays were quantified as previously described (*Zaug et al., 2013*), except the summed counts incorporated into extension products were normalized to both the hTERT levels (determined by western blot) and the loading control and then normalized to WT telomerase. Processivity was measured using the decay method previously described (*Latrick and Cech, 2010*). Briefly, counts for repeats 1–20 were corrected for the number of dGTP nucleotides incorporated. The natural log of the counts left behind in each repeat was graphed vs repeat number and fit by linear regression. The slopes of the linear regressions were compared and normalized to WT telomerase. RAP stimulation by PT, for direct extension assays with POT-TPP1, was measured in the same manner as processivity except values were normalized to WT telomerase with WT POT1-TPP1 (i.e., RAP stimulation by PT = processivity of mutant telomerase with TPP1/processivity of WT telomerase with WT POT1-TPP1). The processivity in the absence of POT1-TPP1 was not included the calculation, because (1) it did not vary greatly among the mutants (*Table 1*) and (2) it was so small relative to the processivity in the presence of POT1-TTP1 that dividing by this distorted the calculations and led to irreproducibility. An alternative fraction method of quantitation was also used to determine RAP stimulation by PT, calculated as the fraction of counts in bands 9 and above divided

by the total sum of counts incorporated, normalized as described for the decay method. Telomerase from stable cell lines was immuno-purified from identical number of cells and was subjected to direct telomerase assays as previously described (*Cohen and Reddel, 2008*). Activity was determined by dividing the total counts by the loading control and normalized to the activity level of cells overexpressing WT hTERT. Telomerase purified from RRLs was assayed using 10 µl of eluate under identical conditions as telomerase from HEK293T cells.

## Northern blot

Northern blots were carried out as previously described, using 5 separate probes for hTR and 2 probes for RNase P (*Xi and Cech, 2014*).

## Molecular cloning

The mCherry-hTERT vector was generated by restriction cloning of the hTERT gene into a modified pBABE-puro vector (a kind gift from Iain Cheeseman) containing the N-terminal mCherry-tag using SalI/EcoRI. GFP-TPP1-OB-LacI was cloned by restriction cloning of the TPP1-OB-fold domain into a modified pBABE-blast vector (a kind gift from Iain Cheeseman) containing the GFP and LacI sequences using XhoI/EcoRI. All point mutations in hTERT and TPP1 were introduced using the Quickchange II mutagenesis kit (Agilent, Santa Clara, CA). The presence of the mutations was verified by Sanger sequencing.

## Cell culture

All human cell lines were cultured at 37°C, 5% $CO_2$ in growth medium (Dulbecco's modified Eagle medium supplemented with 10% fetal bovine serum, 2 mM glutamax [Life Technologies]), 100 units/ml penicillin, and 100 µg/ml streptomycin. The growth medium for the U2OS 2-6-3 cell line (a kind gift from David Spector) was additionally supplemented with 100 µg/ml hygromycin B to maintain the LacO array.

## Transient transfection of human cells

All transient transfections of human cells were carried out using Lipofectamine 2000 (Life Technologies) according to the instructions of the manufacturer.

## Generation of stable cell lines

Retroviruses were generated by transfection of pBABE vectors encoding mCherry-hTERT variants and a puromycin resistance gene along with a vector encoding the coat protein VSV-G into the packaging cell line 293-GP. Virus-containing supernatants were harvested 72 hr post-transfection, filtered through a 0.22-µm filter, and supplemented with 8 µg/ml polybrene. HeLa cells were incubated with virus-containing supernatant for 16 hr. After infection, the growth medium was replaced with fresh medium containing 1 µg/ml puromycin for mCherry-hTERT constructs and 1 µg/ml blasticidin for TPP-FLAG constructs to select for transduced cells. To ensure maintenance of the transgenes, cell lines were kept under selection with 1 µg/ml puromycin and 1 µg/ml blasticidin (TPP1 rescue only) for the duration of the experiments.

## Immunofluorescence and fluorescence

IF experiments were carried out as previously described (*Nandakumar et al., 2012*), using the following antibodies: mouse monoclonal anti-TRF2 (IMG-124A; 1:500; Imgenex), rabbit polyclonal anti-coilin (sc-32860; 1:100; Santa Cruz, Dallas, TX), rat monoclonal anti-mCherry (M11217; 1:1000; Life Technologies), and rabbit polyclonal anti-RAP1 (NB-100-292; 1:500; Novus Biologicals, Littleton, CO). Secondary antibodies (Life Technologies, Abcam) were pre-absorbed to prevent cross-reactivity between rat and mouse antibodies.

## Telomere length analysis

Telomeric restriction fragment length analysis by Southern blot was carried out as previously described (*Nandakumar et al., 2012*). Briefly, 1.5 µg of genomic DNA extracted from human cell lines was digested with RsaI/HinfI and separated by gel electrophoresis using a 0.8% 1xTBE agarose gel for a total of 1100 V-hours. DNA was transferred onto a Hybond N+ membrane (GE Healthcare) and telomeric restriction fragments were detected using a radiolabeled $(TTAGGG)_4$ probe. To determine mean telomere length, lane intensity profiles were extracted using ImageQuant TL and fit to a Gaussian using Kaleidagraph. The mean of the Gaussian distribution was used as telomere length of the respective sample.

## LacI recruitment assay

Plasmids encoding mCherry-hTERT, hTR, and GFP-TPP1-OB-LacI were co-transfected into U2OS 2-6-3 cells grown on coverslips as described above. Cells were fixed for 5 min in PBS supplemented with 0.4% formaldehyde followed by permeabilization with PBS supplemented with 0.2% (vol/vol) Triton X-100 for 5 min. Finally, coverslips were embedded in DAPI containing mounting media (Vector Laboratories, Burlingame, CA). Imaging was carried out visualizing the intrinsic fluorescence of the mCherry- and GFP-fusion proteins without additional staining.

## Microscopy

All images were acquired on a Deltavision Core deconvolution microscope (Applied Precision, GE Healthcare) using a 60× 1.42NA PlanApo N (Olympus, Japan) or 100× UPLanSApo 1.4NA (Olympus) objective and a sCMOS camera. 20 Z-sections with 0.2 µm spacing were acquired for each image with identical exposure conditions within each experiment. For analysis, maximum intensity projections were generated and scaled identically. For presentation in figures, representative images were deconvolved (where indicated), followed by generation of maximum intensity projections of 10 Z-sections, which were scaled identically for all experimental conditions.

## Acknowledgements

We thank Iain Cheeseman, Scott Cohen, Joachim Lingner, and David Spector for generously providing experimental reagents, and members of the Cech lab for helpful discussions. J C S is a Merck fellow of the Damon Runyon Cancer Research Foundation (DRG-2169-13). T R C is an investigator of the HHMI. This work was supported by NIH grant R01 GM099705 to T R C.

## Additional information

### Funding

| Funder | Grant reference number | Author |
|---|---|---|
| National Institutes of Health | RO1-GM099705 | Andrew B Dalby, Thomas R Cech |
| Howard Hughes Medical Institute | | Thomas R Cech |
| Damon Runyon Cancer Research Foundation | DRG-2169-13 | Jens C Schmidt |

The funders had no role in study design, data collection and interpretation, or the decision to submit the work for publication.

### Author contributions

JCS, ABD, Conception and design, Acquisition of data, Analysis and interpretation of data, Drafting or revising the article; TRC, Conception and design, Drafting or revising the article

## Additional files

### Major dataset

The following previously published dataset was used:

| Author(s) | Year | Dataset title | Dataset ID and/or URL | Database, license, and accessibility information |
|---|---|---|---|---|
| Jacobs SA, Podell ER, Cech TR | 2006 | Crystal Structure of the TEN domain of the Telomerase Reverse Transcriptase | http://www.pdb.org/pdb/explore/explore.do?structureId=2B2A | Publicly available at RCSB Protein Data Bank. |

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
