## [Decision Letter]

Thank you for sending your work entitled “Identification of human TERT elements necessary for telomerase recruitment to telomeres” for consideration at *eLife.* Your article has been evaluated by James Manley (Senior editor) and 3 reviewers, one of whom is a member of our Board of Reviewing Editors.

The Reviewing editor and the other reviewers discussed their comments before we reached this decision, and the Reviewing editor has assembled the following comments to help you prepare a revised submission.

In this manuscript the authors carry out a number of experiments to examine the interaction of the TEN domain of TERT with the OB fold domain of TPP1. They present in vivo experiments to document the interaction and specificity. They identify two residues in the TEN domain, K78 and R132 that affect telomerase RAP processivity as well as some effects on catalytic activity and telomere length. They then go on to identify one mutation in TPP1, E215 that compensates for the K78 mutant in the TERT TEN domain. The authors conclude that unlike in S. pombe, where TERT and the TPP1 homologue are bridged by a protein, CCQ, in human cells, these two proteins interact directly. Unfortunately all of the evidence presented here is in vivo and thus no clear conclusion about any possible bridging protein can be made. Rather than directly addressing the potential interaction of the TEN domain and TPP1 by using biochemical assays, the authors used a series of in vivo surrogates that suffer from a number of caveats. For example, high-level protein overexpression is used to examine cellular localization, when over expression is well known to distort protein localization. Without any direct demonstration of an interaction of TPP1 and the TEN domain that is mediated by the charge swap mutants that are described, the conclusions here are not supported by the data.

Major concerns:

If the authors would like to conclude that there is a direct interaction of hTERT TEN domain with TPP1 OB fold evidence for direct protein interacts needs to be presented. The hypothesis they are trying to test is whether these two proteins interact and if that interaction is mediated by the charged residues on the TEL patch of TPP1 and the opposite charges on the TERT- TEN domain. There are a number of biochemical assays that they could use. They could purify the specific proteins or the domains and test interaction and affinities on Biacore chips. They could use GST pull down experiments, affinity columns or similar techniques. This assays presented here including the Cajal body localization and LacO array recruitment are very indirect and are in vivo. Thus they do not exclude a bridging protein, as the authors try to conclude. For the authors to conclude that the interaction of TERT and TPP1 is a direct interaction, and mediated by the charged residues they identify, evidence of direct protein interaction in vitro needs to be presented.

Specific comments:

1) Figure 1: the diagram has blue highlighted text indicating conserved residues; however additional mutants that are not in blue are shown in the Figure 1 supplement. There needs to be an explanation for how specific residues were chosen for mutation and need to state in text, all of the mutants that were made and why. How the mutant combinations in in Table 1 were chosen should be described.

2) The results shown in Figure 1 and Table 1 are essential for understanding the rest of the paper. There are a number of issues in these two figures that need to be clarified. In Figure 1 the authors set up the activity assay and the mutants that are the basis for all of their experiments going forward. It is important to explain in the text what these assays are, and what is concluded from them. I am assuming that the intrinsic changes in processivity are being ignored (Figure 1) and that the PT stimulation is being used as a measure of the interaction of TEN 1 and TPP1. However this is never stated directly. The authors need to explain how they define processivity and PT stimulation differently and how it is quantitated.

This data is quantitated differently from a previous report by the Cech lab in Latrick et. al 2010. The authors should provide an explanation for changing the processivity analysis from previous work, since choosing to sum all counts above band #9 seems a bit arbitrary and less rigorous than previous studies. In Figure 1, there is a clear increased processivity of R132E and K78E+ R132E. This 2-fold increase is not evident from the gel in Figure 1. And also it is not commented on in the paper. If this is true and not due to a normalization artifact, how does this change in processivity affect telomere length? This increased processivity is also reported in Table 1 as 186%, a large effect.

Currently there is very little description on how the authors determine the values they report on Table 1 for telomerase activity, RAP, and RAP stimulation by POT1-TPP1. Since IP-efficiencies and/or expression of hTERT vary greatly among different hTERT mutants, it was often hard to see the correlation between what is shown on gels vs. values in Table 1. Thus, it would be very important for the authors to better describe their quantification methods, and provide raw data before normalization. Were these measurements done on serial dilutions of the extract to increase the accuracy? How reproducible are these assays? How many experiments were done to generate are in the error bars shown? Figure1 and Table 1 describe some of the same mutants. Figure 1 axis label says “normalized PT stimulation” while Table 1 column 4 says “% RAP stimulation”. I assume these are the same thing? Need to use consistent terminology if the reader is to follow.

The authors say that K78E and R132E are required for RAP stimulation but not essential for enzyme activity, however the double mutant has only 63 % activity. This is overstating the conclusion that these residues are not needed for activity and they need to make conclusions that represent the data presented.

3) The overexpression studies need to be cautiously interpreted. To determine if the TERT TEN domain mutants localize to telomeres, the authors overexpress the mutants and examined localization to telomeres or Cajal bodies. This experiment is not sufficient to conclude the proteins fail to localize to telomeres. Overexpression of proteins is known to cause spurious cellular localization. The fact that overexpression of the wt gives different pattern than overexpression of the mutant only indicates that the mutants may aggregate differently in the cell. If they want do make conclusion about protein localization, a second method without overexpression is needed. The authors state they have an antibody to TERT, so ChIP and Co-IP might be an alternative. Finally the authors do not show the level of over expression or look at the stoichiometry of the various holoenzyme components. The authors conclusion from this experiment that TEN domain and and TPP interact is not supported by these experiments. It is not clear this experiment is helpful to the conclusions given the caveats of over expression.

4) It is not clear how the authors are measuring telomerase activity, as it does not seem to be consistent. In Figure 1 the K78E+R132E mutant is reported to have 60 % telomerase activity but here Figure 4 the activity is more than wildtype. I wonder how these numbers are normalized to give these values. It would be much better to show a titration activity with amounts normalized by hTERT levels. The conclusion in the text is that there is no effect of the mutants on activity, but an effect on telomere length. This conclusion is not supported by the data given the results of reduced activity in Figure 1.

5) The method used to show localization of telomerase does not have any controls. In Figure 4 the authors show a robust signal for endogenous hTR localization in cells. Given that there is a copy number of about 100 molecules of hTR per cell, it is not clear what could be generating such a robust signal. To know that this technique can indeed localize endogenous hTR, the authors need a control experiment. They should use a cell line that does not express hTR such as VA13 and show there is no signal with this technique, or use mouse cell lines that are deleted for the endogenous mTR and show there is no signal. Unless a control for specificity is shown this experiment should be omitted. The authors conclude from this experiment that mutant hTERT exerts a dominant effect on endogenous hTR localization. This conclusion is not justified by the data.

6) Figure 4. In all of the previous experiments the authors overexpressed both hTERT and hTR, in this experiment they only overexpressed hTERT; it is not clear why this was done. Given that the mutants K78E and R132E showed decreased telomerase enzyme activity in the experiments in Figure 1, it is not surprising that there is less telomere elongation seen with these overexpressed mutants. Over expression of these mutant enzymes that have decreased activity will sequester all of the hTR and result in overall reduced telomerase activity and thus explain reduced telomere elongation. The authors' conclusion that this data shows that the TEN domain mutants fail to localize but are active is not warranted by the data presented.

7) The authors describe a charge swap experiment in which they mutate residues in TPP1 that they think might interact with the TEN domain acidic residues in Figure 5. The authors state that they mutated “a number of residues in TPP1”, however they then just show E169K and E215K. It is important for the authors to list all the mutants tested and show us what the phenotypes are for all of them, otherwise the specificity of these chosen mutants that will be followed going forward is not clear.

8) The authors use an assay developed by Zhong et al to examine ectopic localization of Tert to a chromosome locus where the OB fold domain of TPP1 has been tethered in Figure 6. They use this assay to demonstrate that mutants in the TEN domains disrupt localization but the double mutant of TEN K78E with TPP1 E215K has an increased number of foci. The figure shown in Figure 6 is not helpful. This figure only shows single cell nuclei and the point of the assay is to examine the number of nuclei with foci. So it would only make sense to show a field of cells, not single cells. So either do not show Figure 6 or show a field that would represent the data they are trying to relate. More importantly since this is an in vivo assay, if there were a bridging protein it would still be present, so the main conclusion about a charged interaction is being the main interaction cannot be concluded from this experiment.

9) The authors add analysis of V144 at the end of the paper in Figure 7. This figure does not contribute in any way to the point of this paper. The V-144 mutation may be of interest in telomerase function in IPF, but the addition of this mutant onto the end of this paper is distracting and does not belong here. This data is best published in another study and should be removed from this manuscript.

[Editors' note: further revisions were requested prior to acceptance, as described below.]

Thank you for resubmitting your work entitled “Identification of human TERT elements necessary for telomerase recruitment to telomeres” for further consideration at *eLife*. Your revised article has been favorably evaluated by James Manley (Senior editor) and a member of the Board of Reviewing Editors. The manuscript has been improved but there are some issues that need to be addressed with further revisions, as outlined in the following review.

The rebuttal letter by the authors is persuasive in many areas, and a number of helpful changes have been added to the manuscript. However some of the changes and additions are confusing and perhaps some figures are mislabeled. If the authors can clarify and revise the writing of the manuscript, it is in principal acceptable for publication.

Here are some of the concerns, listed sequentially as they appear in the manuscript, not in order of importance (some are minor suggestions).

1) In the rebuttal letter the authors argue that it is not possible to look at direct protein interaction with hTERT and TPP1 because of solubility issues that many in the field acknowledge. I would argue that interactions of soluble proteins made in RRL using specific antibodies or GST fusions could be used to look at pull downs and protein or domain interactions. However, I will not push this point too far. The authors are persuasive in the rebuttal that the PT stimulation is an excellent proxy for direct protein-protein interaction' They state in the letter “we consider the charge-swap mutants to provide such a strong argument for a direct protein-protein interaction”

However, in the text of the manuscript the authors did not make a strong case for how the PT stimulation assay is being used as a proxy for protein-protein interactions. Instead the text was altered in the opposite direction. They have removed the wording that stated they were assaying RAP stimulation and instead now say they are measuring protein interactions. I think it is important for the reader to understand the actual experiments that are done measure RAP stimulation, and then at the end of the paragraph the authors can say that they interpret this as a direct interaction and they will go on to test this.

Further on in the text, the authors are again mixing the descriptions of the results with their interpretation. The previous text that is marked as deleted gives a more straightforward description of that was measured and is interpretation from that result. A reversion to this kind of language would help the manuscript significantly.

2) The yellow color in Figure 2 is very difficult to see, I suggest using a different color.

3) The authors use the jargon “Super telomerase” in some cases in the text but not constantly. They are referring to a term used in a paper by Lingner's group when hTERT and hTR are co-overexpressed. The authors use “super telomerase” without explaining what it means. In the methods section the authors state they use the direct telomerase assay described by Zaug 2013. In that paper both hTERT and hTR are co-overexpressed but the term “Super telomerase” is not used. Since the Zaug 2013 assay is used in Figure 1, technically this could be called super telomerase. I suggest that the term be eliminated and the experiment just described as they are. (Note Figure 5—figure supplement 3 also uses the term super telomerase.)

4) In Figure 4, it will be important to show that it is the lack of interaction of TERT with TPP1 that is responsible for the short telomeres and not the decreased enzyme activity intrinsic in the hTERT mutants. I made this point before; however, the authors disagreed in their rebuttal. Figure 4 shows activity assays from cell lines with different levels of telomerase expression. The authors argue that the activity per cell for each of the mutants is the same. And the quantitation below the lanes suggests that the over expressed hTERT mutants have slightly more activity than overexpressed while type. This is discordant with the careful quantitation shown in Figure 1 that shows a decreased catalytic activity of the 78 and 123 mutants.

In their rebuttal the authors state:

“The results in Figure 4 demonstrate that all cell lines over-expressing the hTERT alleles have a 3-4-fold higher telomerase activity than the parental HeLa cell line. This observation is crucial since it rules out lower telomerase activity as a possible explanation for the reduced telomere length in the cell lines expressing hTERT with mutations in the TEN-domain. To the contrary, telomeres shrink in cell-lines expressing TEN-domain mutant alleles despite 3-4-fold higher telomerase activity per cell.”

I am not sure I follow the logic of comparison of the level of activity in untransfected HeLa cells. As I understand it, the comparison here is not with the untransfected HeLa cell telomere length, but between the cell lines that are overexpressing either Wt or various hTERT mutants. The telomere length of untransfected HeLa cells over the 8-week period shown in lanes 1 and 2 are not changed, as expected. When Wt hTERT is overexpressed telomeres clearly lengthen. When the mutants are expressed telomeres fail to lengthen. Given the intrinsic heterogeneity of the telomeres, and following just one clone, is not possible to make accurate comparisons of the various lengths in cell lines where there is a failure to elongate. But it is clear that each of the three mutant analyzed fail to elongate the telomeres while overexpression of wt hTERT does show elongation. In Figure 1 the intrinsic telomerase activity of the mutants were all shown to be reduces and in fact the 78;132 mutant is 60% that of wt. (Why this is not reflected in their quantitation in Figure 4 is not clear, however this in vivo assay may not be as sensitive as the direct assay shown in Figure 1, so I am assuming the discrepancy is due to the inherent experimental error in this in vivo analysis from clones of cells). My question is simply what would happen if an hTERT mutant that is WT in the TEN domains but has 60% catalytic activity were assayed in this manner? Would there be elongation?

The experiments shown in the new Figure 7 go a long way toward solving the problem in Figure 4. Given that there are a number of other issues with Figure 7 (see below), I would suggest that to put this issue to rest the, the authors might find a way to combine Figure 4 with Figure 7 in one figure and not have to resort to arguments over the intrinsic 60% activity of the mutant telomerase.

The experiments to show localization of hTR to telomeres in Figure 4 are problematic and probably not necessary for the conclusions of this paper. I suggest eliminating Figure 4. The authors show a control in Figure 4–figure supplement 1 that suggests this method to examine hTR might have serious technical issues. The HeLa cells shown in Figure 4–figure supplement 1 show that almost all of the green dots do not co-localize with TRF2 or with Colin. The authors statement: 'Although our FISH method readily detected hTR foci that co-localized with telomeres in telomerase-positive HeLa cells, hTR signals were completely absent in telomerase negative VA13 and U2OS cell lines, confirming that FISH is a valid approach to determine the subcellular localization of hTR' is not supported by the data shown in the figure. In the HeLa cells there are 6-8 large green dots and many small green dots. In the Va13 and U2OS the large green dots are not seen but just as many small green dots appear. Given that the there is only one hTR per active telomerase, and the signal would be expected to be quite faint, it is not clear what the large green dots that do not co-localize that are seen in the HeLa cells represent.

Eliminating this experiment from the paper would not alter the conclusion and could strengthen the paper.

5) Figure 5 shows the charge swap experiments. The direct assay data for the hTERT K78E TPP1 E215K combination is compelling. However the authors acknowledge that the R132E mutant does not show much compensation. It is not clear why they this is still shown in the model at the end in Figure 8 as being involved with a charge swap, when it might be some other kind of interaction.

6) In Figure 5—figure supplement 1 there seems to be is a problem with labeling of the gel, perhaps? Why is the “no PT” in this experiment more processive than the WT PT? This higher processivity of no PT is not seen in Figure 5, Figure 5—figure supplement 3 or Figure 5—figure supplement 3.

7) (As a side note, I do not know how supplemental figures should be referred to; it seems confusing to have multiple supplemental Figure 5s. Also some of the figures could be combined or eliminated; not sure why you need Figure 5—figure supplement 1 when that same data shown later; the authors should tighten up their use of supplemental figures.)

8) Figure 6 is now out of order, the new experiments in Figure 7 relate to what is shown in Figures 4 and 5, while Figure 6 is something different.

The variability of the number mCh-hTERT dots in the nucleus of both of Figure 6 and Figure 6—figure supplement 1 is very confusing. In the top panel of Figure 6 only one dot is shown for mCh-hTERT. This would suggest hTERT is not at normal telomeres but rather only at the lacO array. However in the lower panel with the E215TPP1 mutant, not the Wildtype hTERT is at many loci. One way to interpret that is that in this mutant now the hTERT goes to telomeres. But of course this is the opposite of the authors' conclusion. What do all those other dots represent? Do they co localize to telomeres? These experiments are very difficult to interpret. Was this quantitation shown in Figure 6 done in in a blinded fashion? Best practice for this kind of experiment is to have the person scoring the localization blind to the genotype of the cells being examined. Given the noise in this experiment, it would be a good idea to go back to the images and blind them and redo the statistics with the blinked data. The figure legend would then reflect that this experiment was carried out using best practices for such image analysis.

9) Figure 7 is equally difficult to interpret; I suggest eliminating it and merging Figure 7 with the Southern blot in Figure 4.

In the images in this figure most of the hTR signal does not co-localize with the Rap1. (Why was Rap1 used here when the previous figure used TRF2?) Was this quantitation carries out in a blinded fashion? If I understand the numbers in white on the final column, cells expressing K78E hTERT and WT TPP1 show 49/52 or 94% co-localization, whilst cells expressing K78E hTERT and E215K TPP1 show 20/51 or 40% co-localization. Isn't this the opposite of what you would expect? This experiment needs to be redone in a blinded fashion, or simply eliminate these co-localization experiments that are problematic.

10) The model of TPP1 interaction with TERT is shown in Figure 8 and also in Figure 5. It is not clear that it needs to be in both places. The way that TPP1 is drawn here however might confuse people. The three domains should not have black outlines around them; it makes it look like this is three Different proteins. The color difference between TPP1 and TIN2 is subtle, so hard to know which protein the reddish colors circle belongs to. Eliminating the black line will help this. Similarly if you eliminate the black outline around the TEN domains where it touches the TERT, it will look much more like a domain and not a separate protein.

[Editors' note: further clarifications were requested prior to acceptance, as described below.]

Thank you for resubmitting your work entitled “Identification of human TERT elements necessary for telomerase recruitment to telomeres” for further consideration at *eLife.* Your revised article has been favorably evaluated by James Manley (Senior editor) and a member of the Board of Reviewing Editors. The manuscript has been improved but there are some remaining issues that need to be addressed before acceptance, as outlined below:

In the revised manuscript the authors address a number of concerns. However the authors have not addressed two major points that were raised.

First the evidence shown in Figure 4 is not sufficient to say that “telomeres shrink” when the TEN domain mutants are overexpressed. The heterogeneity in the telomere lengths is just too variable. It is clear that telomere do not elongate as they do when the Wt TERT is overexpressed. To convince the readers that there is actually significant shortening of telomeres, with these heterogeneous lengths, it would be necessary to show several independent clones and show a decrease occurs reproducibly in several independent clones. It is well established that there is significant telomere heterogeneity in culture, so one sample that appears marginally shorter in one lane, without repetition is not convincing. I understand redoing this experiment is a lot of work. For that reason it would be best to simply narrow the conclusion to what can clearly be established from this data: that the TEN domain mutants fail to elongate telomeres like the Wt TERT does. My earlier comment that the double mutant has less activity than the WT TERT is still a concern. This decreased activity is not shown in this figure but is shown in Figure 1 where more quantitative activity assays are done.

The second point is that the hTR localization to telomeres is expected to be a very difficult thing to measure. It is clear that hTR localization to cajal bodies can be established, and it is expected that many hTR molecules would be localized in this processing center, however it has not been clearly established that a single telomerase at a telomere can be detected by this technique.

The authors state, “Note that the FISH method we are using in this study is well established in the literature and has been used by other labs, including the Artandi and Terns labs.” Just because others use an assay, does not make it robust. For example many labs do TRAP assays to quantitate telomerase activity using 40 or more rounds of PCR; I am sure the Cech lab appreciates that while widely used, that TRAP assay is not quantitative. None of the prior publications on hTR FISH that the authors refer to did controls to show their hybridization detected single telomerase molecules at telomeres. Were mutant oligos that are mismatched still giving a signal in this assay? What to the foci that do not co localize with telomeres represent? What do the few foci of hTR hybridization represent given that that most telomeres show no signal? This lack of signal at most telomeres was even more apparent in the previous Figure 7 that the authors decided to remove.

Do the authors conclude that a signal at a telomere mean they are visualizing ONE telomerase RNA molecule localized to a telomere? Is it possible to detect just three the fluorescent probe oligos under these conditions? The TRF2 signal clearly comes from many hundreds of TRF2 proteins bound to the telomere, and yet the foci of hTR signal are bright. Have the authors considered that this may be an aggregate of hTR? It would be helpful for these authors to state directly if they think they are measuring a cluster of hTR molecules or if that they believe they are seeing one telomerase molecule on one telomere, with their microscopy. If they authors conclude there may be multiple hTR molecules in these foci, it would be helpful to explain why there would be multiple telomerase molecules and how that affects the charge swap interaction they are proposing as shown in Figure 8.

The uncertainties in these experiments weaken the manuscript. The conclusion on processivity can be made without this FISH data.

One final note it was surprising to see the authors state in the rebuttal that under some salt conditions TERT alone is more processive than TERT + Pot1/Tpp1. They say “Figure 5—figure supplement 1 is labeled correctly. The gel was a screen carried out under non-physiological salt conditions, and under these conditions telomerase alone is more processive.” This reduced my confidence in the biological result of the Pot1/Tpp1 stimulation. Why is the effect so sensitive to salt conditions? It is not clear what “non-physiological salt conditions” is and why they would have used these conditions in the experiment shown in Figure 5—figure supplement 1. Presenting this figure will be very confusing to the readers and it is not clear why it is needed. If the salt conditions are this sensitive such that the exact opposite of the conclusion the authors are making can easily be obtained, it is important to state this in the methods, and not just refer to previous papers for the details. For others to be able to reproduce the results, understanding the salt issues is important.

---

## [Author Response]

We thank you for your detailed comments, which we’ve taken very seriously in our attached revision. In several cases we present new experiments: new Figure 3 in response to request to quantify expression levels, a Figure 4–figure supplement 1 showing the requested controls for hTR FISH using telomerase-negative VA13 and U2OS cell lines, the requested test of “direct binding” by expressing telomerase in RRL’s in a new Figure 5—figure supplement 3, and a new Figure 7 showing dramatic rescue of telomerase recruitment and telomere length extension in vivo with the charge-swap mutant hTERT and TPP1 combination. The requested re-analysis of enzyme processivity using a second method is presented in Figure 1—figure supplement 2. Finally, the writing has been clarified throughout, and a revised cartoon in Figure 5 should make it easier for the reader to understand why we consider the charge-swap mutants to provide such a strong argument for a direct protein-protein interaction, especially in light of other mutants at non-interacting sites which show the opposite effect.

*Major concerns*:

*If the authors would like to conclude that there is a direct interaction of hTERT TEN domain with TPP1 OB fold evidence for direct protein interacts needs to be presented. The hypothesis they are trying to test is whether these two proteins interact and if that interaction is mediated by the charged residues on the TEL patch of TPP1 and the opposite charges on the TERT- TEN domain. There are a number of biochemical assays that they could use. They could purify the specific proteins or the domains and test interaction and affinities on Biacore chips. They could use GST pull down experiments, affinity columns or similar techniques. This assays presented here including the Cajal body localization and LacO array recruitment are very indirect and are in vivo. Thus they do not exclude a bridging protein, as the authors try to conclude. For the authors to conclude that the interaction of TERT and TPP1 is a direct interaction, and mediated by the charged residues they identify, evidence of direct protein interaction in vitro needs to be presented*.

Purification of telomerase in quantities large enough for biochemical binding assays, such as the ones suggested by the reviewers, has not been accomplished in any lab. The data presented in this study include biochemical assays testing the interaction between TPP1 and telomerase *in vitro*. It is well established in the literature that the stimulation of telomerase processivity *in vitro* by POT1-TPP1 is due to an interaction between the TEL-patch of TPP1 and telomerase ([30]; [37]; Zaug et al, 2010). We carry out these assays with recombinant POT1-TPP1 complex, purified from insect cells and bacteria respectively, and telomerase immuno-purified from HEK293T using an affinity tag and protease mediated elution after stringent washing steps. Therefore, the only way a potential bridging factor could still be present is if it were to be tightly associated with telomerase. To address the reviewer’s concerns about a co-purifying bridging factor, we have repeated the *in vitro* charge swap experiments with telomerase produced and purified from rabbit reticulocyte lysates (RRL). The results with obtained using RRL telomerase are consistent with those observed with immuno-purified super-telomerase, with statistically significant differences. In addition we have also included new data, demonstrating that E215K TPP1 restores telomeric localization and telomere maintenance in cells. In total we now demonstrate the charge rescue using multiple independent experiments *in vivo* and telomerase from two sources *in vitro*. In addition we would like to point out that a compensatory effect of combining two individually deleterious mutations is inconsistent with the presence of a bridging factor. We have now illustrated this concept in more detail in Figure 5 and provided several additional examples from the literature in the Discussion, which use charge-swap experiments to assess direct protein-protein interactions.

*Specific comments*:

*1)*
Figure 1*: the diagram has blue highlighted text indicating conserved residues; however additional mutants that are not in blue are shown in the*
Figure 1
*supplement. There needs to be an explanation for how specific residues were chosen for mutation and need to state in text, all of the mutants that were made and why. How the mutant combinations in in*
Table 1
*were chosen should be described*.

We have modified the text to further clarify our rationale for choosing basic residues in the TEN domain . Without *a priori* knowledge of which hTERT residues interact with TEL-patch, we chose to mutate as many basic residues as we reasonably could. Mutants with multiple mutations were made to screen TEN domain residues more efficiently, or to examine the compound effects of single mutations.

*2) The results shown in*
Figure 1
*and*
Table 1
*are essential for understanding the rest of the paper. There are a number of issues in these two figures that need to be clarified. In*
Figure 1
*the authors set up the activity assay and the mutants that are the basis for all of their experiments going forward. It is important to explain in the text what these assays are, and what is concluded from them*.

We thank the reviewers for their comments; these issues have been corrected in the text of the manuscript.

*I am assuming that the intrinsic changes in processivity are being ignored (*Figure 1*)*…

Correct. In addition, the intrinsic processivity differences were influenced by a normalization artefact. We have re-calculated processivity as per the reviewer suggestion, and the differences are now minimal.

*…and that the PT stimulation is being used as a measure of the interaction of TEN 1 and TPP1. However this is never stated directly*.

Indeed PT stimulation is used as a measure of the interaction between TEN and TPP1, we have further clarified this in the text.

*The authors need to explain how they define processivity and PT stimulation differently and how it is quantitated*.

We have also clarified this within the text and Materials and methods.

*This data is quantitated differently from a previous report by the Cech lab in Latrick et. al 2010. The authors should provide an explanation for changing the processivity analysis from previous work, since choosing to sum all counts above band #9 seems a bit arbitrary and less rigorous than previous studies*.

We now present processivity and RAP stimulation by PT values made using the decay method described by Latrick and Cech, so Table 1, Figure 1, and Figure 5 are updated. In addition we compare the +9 (fractional) and decay methods directly Figure 1—figure supplement 2, Figure 5—figure supplement 2. Importantly, both methods of quantitation provide similar numbers as observed in previous comparisons (30; 52).

*In*
Figure 1*, there is a clear increased processivity of R132E and K78E+ R132E. This 2-fold increase is not evident from the gel in*
Figure 1*. And also it is not commented on in the paper. If this is true and not due to a normalization artifact, how does this change in processivity affect telomere length? This increased processivity is also reported in*
Table 1
*as 186%, a large effect*.

We thank the reviewers for the suggestion; the large processivity effects were influenced by normalization and have now been corrected.

*Currently there is very little description on how the authors determine the values they report on*
Table 1
*for telomerase activity, RAP, and RAP stimulation by POT1-TPP1*.

We have further clarified our description in materials and methods, as well as in the table and in the text itself (and Materials and methods).

*Since IP-efficiencies and/or expression of hTERT vary greatly among different hTERT mutants, it was often hard to see the correlation between what is shown on gels vs. values in*
Table 1*. Thus, it would be very important for the authors to better describe their quantification methods*, *and provide raw data before normalization. Were these measurements done on serial dilutions of the extract to increase the accuracy?*

We have further clarified the quantification of telomerase enzyme assays in the Materials and methods and Table 1. The purpose of supplement to

Figure 1 is to show the raw data for a variety of hTERT mutants that were screened in order to identify separation-of-function mutants. The mutagenesis and production of super-telomerase is technically challenging, expensive, and time consuming. For this reason the mutants were produced in small groups at different times, which can result in expression variability. We controlled for this by simultaneously producing wild-type telomerase with each group of mutants. As the purpose of the screen was to identify separation-of-function mutants, we did not further characterize many of the mutants from the screen.

The data in supplement to Figure 1 and in Table 1 show that we screened a variety of mutants. In contrast, Figure 1 presents data that is integral to the main conclusions of the paper. For Figure 1, the production of wild-type and mutant super-telomerases were carried out simultaneously, and in this case the hTERT expression levels were quite similar, see Figure 1. Although we do not typically do serial dilutions, the telomerase assays were done under standard Cech Lab conditions, and the linearity of our measurements with respect to concentration and time have been fully documented in of a previous report (52).

*How reproducible are these assays? How many experiments were done to generate are in the error bars shown*?

The number of experimental replicates is listed for all *in vitro* telomerase assays as well as *in vivo* cell-based experiments, in each figure legend or detailed in the table. The error bars on the graphs or standard deviations noted in Table 1 give a measure of the reproducibility of the assays*.*

*Figure1 and*
Table 1
*describe some of the same mutants.*
Figure 1
*axis label says “normalized PT stimulation” while*
Table 1
*column 4 says “% RAP stimulation”. I assume these are the same thing? Need to use consistent terminology if the reader is to follow*.

Yes. This oversight on our part has been corrected.

*The authors say that K78E and R132E are required for RAP stimulation but not essential for enzyme activity, however the double mutant has only 63 % activity. This is overstating the conclusion that these residues are not needed for activity and they need to make conclusions that represent the data presented*.

The text has been modified to more accurately emphasize the point we intended to make. K78E hTERT has more than 90% of wild-type activity and intrinsic processivity, and all experiments testing the direct nature of the hTERT-TPP1 are done using this allele.

*3) The overexpression studies need to be cautiously interpreted. To determine if the TERT TEN domain mutants localize to telomeres, the authors overexpress the mutants and examined localization to telomeres or Cajal bodies. This experiment is not sufficient to conclude the proteins fail to localize to telomeres. Overexpression of proteins is known to cause spurious cellular localization. The fact that overexpression of the wt gives different pattern than overexpression of the mutant only indicates that the mutants may aggregate differently in the cell*.

In terms of overexpression (OE) causing aggregation, the catalytic activity of telomerase is very sensitive to the integrity of this RNP complex; our careful measurements of the activity (Vmax and Km) of the overexpressed and endogenous telomerases show that they have the same activity per molecule (48), arguing against aggregation upon OE. Furthermore, the measurements herein show that key mutants retain WT-like activity, arguing against differential aggregation. And while we agree with the reviewers that protein OE can in some cases lead to abnormal sub-cellular localization of proteins, our mutants do not form inclusion bodies or localize to non-physiological structures. Instead, our TEN-domain mutants fail to localize to telomeres despite OE, and remain sequestered in Cajal bodies. Since TEN-domain mutants localize to Cajal bodies and not some other structure, there is no reason to believe that there is an assembly or maturation defect caused by overexpression. We believe the fact that TEN-domain mutants can’t be forced to localize to telomeres even when over-expressed strengthens the conclusion that they fail to interact with TPP1. We would also like to point out that over-expression of telomerase has been previously used to analyse its subcellular localization, including the relevance of hTERT residues on this process (13; 39; 54).

*If they want do make conclusion about protein localization, a second method without overexpression is needed. The authors state they have an antibody to TERT, so ChIP and Co-IP might be an alternative*.

We appreciate the reviewers’ comment, but the only approach to avoid over-expression is genome editing, which is far beyond the scope of this study. We would like to point out that we also tested telomerase localization by hTR FISH under conditions where only hTERT is over-expressed, which only raised telomerase activity per cell about 3-4-fold, compared to the ∼200-fold in super-telomerase cells (13). Furthermore, our LacO experiments in Figure 6 demonstrate that K78E telomerase is fully capable of associating with the OB-fold domain of TPP1 when the E215K mutation is introduced. Finally, we have now included experiments showing that introduction of E215K TPP1, but not WT or E169K TPP1, rescues telomere localization of K78E telomerase, adding further support for our model (Figure 7).

*Finally the authors do not show the level of over expression or look at the stoichiometry of the various holoenzyme components. The authors conclusion from this experiment that TEN domain and and TPP interact is not supported by these experiments. It is not clear this experiment is helpful to the conclusions given the caveats of over expression*.

We have now included western and northern blots that demonstrate that hTERT and hTR are over-expressed to similar levels under all conditions, ruling out the possibility that the phenotypes observed are due to variable expression levels of the holoenzyme components (Figure 3).

*4) It is not clear how the authors are measuring telomerase activity, as it does not seem to be consistent. In*
Figure 1
*the K78E+R132E mutant is reported to have 60 % telomerase activity but here*
Figure 4
*the activity is more than wildtype. I wonder how these numbers are normalized to give these values. It would be much better to show a titration activity with amounts normalized by hTERT levels. The conclusion in the text is that there is no effect of the mutants on activity, but an effect on telomere length. This conclusion is not supported by the data given the results of reduced activity in*
Figure 1.

Thank you for this comment. We were not clear in the text about the differences between the experiments in Figures 1 and 4, and have proceeded to point out the specific differences and conclusion in the revised text. In Figure 1 we analyse the activity of telomerase over-expressed in HEK293T cells followed by immuno-purification using the ProA-tag on the N-terminus of hTERT. The activity levels in the experiments in Figure 1 are normalized to both the loading control and the intensity of the western blot band shown. The activity shown in Figure 4 is a measure of telomerase activity per cell and is normalized to both loading control and the number of cells used to purify telomerase. Since endogenous telomerase lacks an affinity tag we had to use an alternative approach to purify telomerase from the cell lines stably expressing the different hTERT alleles and the parental HeLa cell line. We chose to use a well-established method using a polyclonal antibody against hTERT to immuno-purify telomerase from these cell lines (11). The results in Figure 4 demonstrate that all cell lines over-expressing the hTERT alleles have a 3-4-fold higher telomerase activity than the parental HeLa cell line. This observation is crucial since it rules out lower telomerase activity as a possible explanation for the reduced telomere length in the cell lines expressing hTERT with mutations in the TEN-domain. To the contrary, telomeres shrink in cell-lines expressing TEN-domain mutant alleles despite 3-4-fold higher telomerase activity per cell. This observation in combination with the lack of telomerase present at telomeres in these cells, lead us to the conclusion that telomeres shrink due to the failure of telomerase association with telomeres. The differences in activity observed could be due to the experimental set up or purification procedure, but they don’t compromise the main conclusions of either figure: K78 and R132 are involved in the interaction between telomerase and TPP1 (and to a lesser degree impact catalytic activity) and thus fail to maintain telomere length due to the failure to localize to telomeres.

*5) The method used to show localization of telomerase does not have any controls. In*
Figure 4
*the authors show a robust signal for endogenous hTR localization in cells. Given that there is a copy number of about 100 molecules of hTR per cell, it is not clear what could be generating such a robust signal. To know that this technique can indeed localize endogenous hTR, the authors need a control experiment. They should use a cell line that does not express hTR such as VA13 and show there is no signal with this technique, or use mouse cell lines that are deleted for the endogenous mTR and show there is no signal. Unless a control for specificity is shown this experiment should be omitted. The authors conclude from this experiment that mutant hTERT exerts a dominant effect on endogenous hTR localization. This conclusion is not justified by the data*.

We have now included VA13 and U2OS ALT cell lines as controls for the FISH experiment (Figure 4–figure supplement 1) and neither cell line has detectable hTR foci at telomeres. Thus the conclusion that over-expression of an hTERT allele, that fails to localize to telomeres, displaces endogenous hTR from telomeres is fully supported by the data presented.

*6)*
Figure 4*. In all of the previous experiments the authors overexpressed both hTERT and hTR, in this experiment they only overexpressed hTERT; it is not clear why this was done*.

To assess telomere length over an extended period of time it was necessary to stably express all hTERT variants in HeLa cells. Importantly, all hTERT variants are overexpressed relative to endogenous telomerase (see western, Figure 4). Therefore, the vast majority of hTR, which is the limiting component for telomerase assembly under these conditions, should be assembled with exogenous hTERT molecules. Additionally, overexpression of only hTERT leads to a 3-4-fold increase in telomerase activity since endogenous hTR is not limiting the level of telomerase, while overexpression of both components leads to a 200-fold increase (13). Therefore, by choosing to only overexpress hTERT, telomerase activity is much more similar to endogenous telomerase activity than if both components were overexpressed. We have now clarified the rationale for the experimental set up in the text.

*Given that the mutants K78E and R132E showed decreased telomerase enzyme activity in the experiments in*
Figure 1*, it is not surprising that there is less telomere elongation seen with these overexpressed mutants. Over expression of these mutant enzymes that have decreased activity will sequester all of the hTR and result in overall reduced telomerase activity and thus explain reduced telomere elongation. The authors' conclusion that this data shows that the TEN domain mutants fail to localize but are active is not warranted by the data presented*.

This concern is not true, and we have clarified the situation in the revision. In fact, all cell lines have at least 3-fold higher telomerase activity per cell, relative to the parental HeLa cell line, due to hTERT overexpression (Figure 4). As described (48), overexpression of hTERT increases the fraction of hTR assembled in telomerase RNPs. We have now clarified this reasoning in the text. The parental HeLa cell line maintains a constant telomere length. In contrast, telomeres shrink in cell lines expressing K78E, R132E, or K78E;R132E hTERT despite ∼3-4-fold higher telomerase activity per cell than the parental HeLa cell line (Figure 4). Additionally, the cell line expressing wild-type hTERT displays significant telomere lengthening with similar telomerase activity per cell compared to all mutant cell lines. Thus, decreased telomerase activity can’t explain telomere shrinking in TEN domain mutant cell lines. Since hTR is sequestered in Cajal bodies in all mutant cell lines, the data fully support the model that telomeres shrink because mutations in the TEN-domain compromise telomerase localization to telomeres.

*7) The authors describe a charge swap experiment in which they mutate residues in TPP1 that they think might interact with the TEN domain acidic residues in*
Figure 5*. The authors state that they mutated “a number of residues in TPP1”, however they then just show E169K and E215K. It is important for the authors to list all the mutants tested and show us what the phenotypes are for all of them, otherwise the specificity of these chosen mutants that will be followed going forward is not clear*.

TPP1 mutant E169K was included as a control to demonstrate specificity of E215K for telomerase K78E. In addition we have added Figure 5—figure supplement 1, including all other residues mutated in TPP1, demonstrating that only E215K compensates for the presence of the K78E mutation in telomerase.

*8) The authors use an assay developed by Zhong et al to examine ectopic localization of Tert to a chromosome locus where the OB fold domain of TPP1 has been tethered in*
Figure 6*. They use this assay to demonstrate that mutants in the TEN domains disrupt localization but the double mutant of TEN K78E with TPP1 E215K has an increased number of foci. The figure shown in*
Figure 6
*is not helpful. This figure only shows single cell nuclei and the point of the assay is to examine the number of nuclei with foci. So it would only make sense to show a field of cells, not single cells. So either do not show*
Figure 6
*or show a field that would represent the data they are trying to relate*.

Thank you for the suggestion. We fully agree and have added additional examples of cells for all panels relevant for the charge-swap as Figure 6—figure supplement 1. Due to the transfection efficiency being less than 100% it is not possible to capture fields with high densities of relevant cells. The images presented in the main figure are representative images of the 117-176 nuclei evaluated per condition in three independent experiments.

*More importantly since this is an in vivo assay, if there were a bridging protein it would still be present, so the main conclusion about a charged interaction is being the main interaction cannot be concluded from this experiment*.

We believe that the LacO-assay presented in Figure 6 fully supports our model that the OB-fold of TPP1 and the TEN-domain must interact directly, via the following argument. Mutation of the TEN-domain or the TEL-patch independently reduces the association between telomerase and the OB-fold domain of TPP1, which is manifested in a reduced fraction of nuclei in which telomerase and the GFP-TPP1 OB-LacO foci co-localize. If each independent mutation reduces the affinity for one or more potential bridging factors, simultaneous mutation of the TEN-domain and the TEL-patch should lead to an additive reduction of the fraction of nuclei that show co-localization of telomerase and OB-fold foci. We observe the opposite result: mutation of the TEL-patch rescues the presence of a mutation in the TEN-domain. While WT-telomerase co-localizes with E215K OB-fold in only 20% of the nuclei, K78E-telomerase co-localizes with E215K OB-fold in over 75% of the nuclei. The only plausible explanation is that mutation of the TEN-domain or TEL-patch independently eliminates a direct interaction between these charged residues, which is re-established by swapping the charged residues on both surfaces. We have now added an illustration to Figure 5 outlining not only how a charge swap can support a direct interaction between the TEN-domain and the TEL-patch, but also how it can rule out the presence of a bridging factor. (See also response to #9 below.)

*9) The authors add analysis of V144 at the end of the paper in*
Figure 7*. This figure does not contribute in any way to the point of this paper. The V-144 mutation may be of interest in telomerase function in IPF, but the addition of this mutant onto the end of this paper is distracting and does not belong here. This data is best published in another study and should be removed from this manuscript*.

We agree with the reviewers that the data on the V144M mutation is out of place at this point in the manuscript, yet it is of interest to explain the molecular basis of this allele causing IPF. We have instead included it as supplement to Figure 5 since it illustrates how mutations in the TEL-patch and the TEN-domain display additive losses in RAP stimulation if they affect residues that do not directly interact, solidifying our *in vitro* analysis of the interaction between telomerase and TPP1.

*[Editors' note: further revisions were requested prior to acceptance, as described below*.*]*

*1) [… In] the text of the manuscript the authors did not make a strong case for how the PT stimulation assay is being used as a proxy for protein-protein interactions. Instead the text was altered in the opposite direction. They have removed the wording that stated they were assaying RAP stimulation and instead now say they are measuring protein interactions*.

This sentence is in the introduction summarizing our conclusions.

*I think it is important for the reader to understand the actual experiments that are done measure RAP stimulation, and then at the end of the paragraph the authors can say that they interpret this as a direct interaction and they will go on to test this*.

*Further on in the text, the authors are again mixing the descriptions of the results with their interpretation. The previous text that is marked as deleted gives a more straightforward description of that was measured and is interpretation from that result. A reversion to this kind of language would help the manuscript significantly*.

We have taken the reviewer’s suggestion and reverted to the original language used to discuss the results at the end of [Sec s1].

*2) The yellow color in*
Figure 2
*is very difficult to see, I suggest using a different color.*

We have revised Figure 2 for better visibility.

*3) The authors use the jargon “Super telomerase” in some cases in the text but not constantly. They are referring to a term used in a paper by Lingner's group when hTERT and hTR are co-overexpressed. The authors use “super telomerase” without explaining what it means. In the methods section the authors state they use the direct telomerase assay described by*
[52]*. In that paper both hTERT and hTR are co-overexpressed but the term “Super telomerase” is not used. Since the*
[52]
*assay is used in*
Figure 1*, technically this could be called super telomerase. I suggest that the term be eliminated and the experiment just described as they are. (Note*
Figure 5—figure supplement 3
*also uses the term super telomerase*.*)*

We have removed the term “super telomerase” from text and figures.

*4) […] In their rebuttal the authors state*:

*“The results in*
Figure 4
*demonstrate that all cell lines over-expressing the hTERT alleles have a 3-4-fold higher telomerase activity than the parental HeLa cell line. This observation is crucial since it rules out lower telomerase activity as a possible explanation for the reduced telomere length in the cell lines expressing hTERT with mutations in the TEN-domain. To the contrary, telomeres shrink in cell-lines expressing TEN-domain mutant alleles despite 3-4-fold higher telomerase activity per cell*.*”*

*I am not sure I follow the logic of comparison of the level of activity in untransfected HeLa cells. […] My question is simply what would happen if an hTERT mutant that is WT in the TEN domains but has 60% catalytic activity were assayed in this manner? Would there be elongation*?

We have sought to clarify this further within the text. We understand the reviewer’s concerns about the differences between the activity levels in Figures 1 and 4. The purification methods are different in these experiments and might be the reason for the differences. There is a higher degree of variability in the data shown in Figure 4, as is reflected in the error bars. The conclusion we are trying to draw from these experiments is that telomeres shrink due to a defect in telomere recruitment of telomerase, not an activity defect. The untransfected control maintains the same telomere length, while overexpression of WT hTERT leads to a significant growth in telomere length, due to a 4-fold increase in telomerase activity per cell compared to the untransfected control. In contrast, overexpression of TEN-domain mutants leads to telomere shortening, even though the telomerase activity per cell is increased to a similar degree as with WT hTERT. Thus, cells expressing TEN-domain mutants have much higher telomerase activity than untransfected cells, yet their telomeres are not maintained at the same level as untransfected cells, therefore an activity defect cannot be the cause of telomere erosion. If the telomere shortening of R132E and K78E;R132E were merely due to an activity defect, overexpression of hTERT and thereby raising the activity level 3-4 fold should compensate for the activity defect, which it does not.

*The experiments shown in the new*
Figure 7
*go a long way toward solving the problem in*
Figure 4*. Given that there are a number of other issues with*
Figure 7
*(see below), I would suggest that to put this issue to rest the, the authors might find a way to combine*
Figure 4
*with*
Figure 7
*in one figure and not have to resort to arguments over the intrinsic 60% activity of the mutant telomerase*.

All conclusions in Figure 7 as well as the charge-swap supporting the direct interaction of telomerase with TPP1 are based on the K78E mutant, which has greater than 90% activity in all assays. Combining Figures 4 and 7 would disrupt the logic of the paper, so we have not incorporated the reviewer’s suggestion in this case.

*The experiments to show localization of hTR to telomeres in*
Figure 4
*are problematic and probably not necessary for the conclusions of this paper. I suggest eliminating*
Figure 4*. The authors show a control in Figure 4–figure supplement 1 that suggests this method to examine hTR might have serious technical issues. The HeLa cells shown in Figure 4–figure supplement 1 show that almost all of the green dots do not co-localize with TRF2 or with Colin. The authors statement: 'Although our FISH method readily detected hTR foci that co-localized with telomeres in telomerase-positive HeLa cells, hTR signals were completely absent in telomerase negative VA13 and U2OS cell lines, confirming that FISH is a valid approach to determine the subcellular localization of hTR' is not supported by the data shown in the figure. In the HeLa cells there are 6-8 large green dots and many small green dots. In the Va13 and U2OS the large green dots are not seen but just as many small green dots appear. Given that the there is only one hTR per active telomerase, and the signal would be expected to be quite faint, it is not clear what the large green dots that do not co-localize that are seen in the HeLa cells represent*.

*Eliminating this experiment from the paper would not alter the conclusion and could strengthen the paper*.

In the FISH control HeLa cells show 10 bright signals that are not present in the VA13 or U2OS cells in addition the background signals that are present in all samples. 9 out of these 10 signals co-localize with either the telomeric or the Cajal body marker, which is the expected localization of hTR. We have clarified the text to accommodate the reviewer’s concern. (Note that the FISH method we are using in this study is well established in the literature and has been used by other labs, including the Artandi and Terns labs. We scaled the images to include some background signal, which is the appropriate practice with microscopy data.)

*5)*
Figure 5
*shows the charge swap experiments. The direct assay data for the hTERT K78E TPP1 E215K combination is compelling. However the authors acknowledge that the R132E mutant does not show much compensation. It is not clear why they this is still shown in the model at the end in*
Figure 8
*as being involved with a charge swap, when it might be some other kind of interaction*.

As noted in the text, mutation to R132 compromises RAP stimulation by PT, suggesting that it interacts with the TPP1 TEL-patch. Although we cannot rescue the defect of R132E with a charge-swapped TPP1, the stimulation defect suggests that it is interacting. We have clarified the text in figure legend 8 to make it clear that the question mark does not represent a salt-bridge.

*6) In*
Figure 5—figure supplement 1
*there seems to be is a problem with labeling of the gel, perhaps? Why is the “no PT” in this experiment more processive than the WT PT? This higher processivity of no PT is not seen in*
Figure 5*,*
Figure 5—figure supplement 3
*or*
Figure 5—figure supplement 3.

Figure 5—figure supplement 1 is labeled correctly. The gel was a screen carried out under non-physiological salt conditions, and under these conditions telomerase alone is more processive. We have clarified this in the figure legend, and note that the charge-swap works under a variety of in vitro conditions.

*7) (As a side note, I do not know how supplemental figures should be referred to; it seems confusing to have multiple supplemental*
Figure 5*s. Also some of the figures could be combined or eliminated; not sure why you need*
Figure 5—figure supplement 1
*when that same data shown later; the authors should tighten up their use of supplemental figures*.*)*

We are following the *eLife* guidelines for supplemental figures. Figure 5—figure supplement 1, as well as the majority of other figure supplements, were included to address reviewer concerns, or requests for additional data.

*8)*
Figure 6
*is now out of order, the new experiments in*
Figure 7
*relate to what is shown in*
Figures 4 and 5*, while*
Figure 6
*is something different*.

The logic of the paper is as follows. We first demonstrate the interaction between the TEN-domain *in vitro* and *in vivo* and then go on to demonstrate that the interaction is direct in Figures 5, 6 and 7. Figure 5 shows the *in vitro* rescue and Figure 6 the *in vivo* rescue. We therefore wish to maintain the figure order as is.

*The variability of the number mCh-hTERT dots in the nucleus of both of*
Figure 6
*and*
Figure 6—figure supplement 1
*is very confusing. In the top panel of*
Figure 6
*only one dot is shown for mCh-hTERT. This would suggest hTERT is not at normal telomeres but rather only at the lacO array. However in the lower panel with the E215TPP1 mutant, not the Wildtype hTERT is at many loci. One way to interpret that is that in this mutant now the hTERT goes to telomeres. But of course this is the opposite of the authors' conclusion. What do all those other dots represent? Do they co localize to telomeres? These experiments are very difficult to interpret. Was this quantitation shown in*
Figure 6
*done in in a blinded fashion? Best practice for this kind of experiment is to have the person scoring the localization blind to the genotype of the cells being examined. Given the noise in this experiment, it would be a good idea to go back to the images and blind them and redo the statistics with the blinked data. The figure legend would then reflect that this experiment was carried out using best practices for such image analysis*.

We have added text to further clarify the lacO experiment. When expressing WT mCh-hTERT alongside E215K TPP1, telomerase cannot interact with the TPP1-OB-LacI fusion, but since it is WT it can localize to endogenous telomeres, as visualized by the large number of foci present under these conditions. When it is expressed alongside WT TPP1 it localizes to the LacI focus since the amount of TPP1 present at this site is far greater than any individual telomere. When telomerase can interact with neither the telomere nor the TPP1-OB fusion, as is the case with R132 and R132/K78E mCh-hTERT, it is diffusely localized in the nucleoplasm. We agree with the reviewer that best practice is to blindly quantify experiments if they involve the judgment of the observer. In this case, however, the results are very clear: there is either a mCherry signal present at the GFP focus or there is not. The experiment was carried out three independent times quantifying a large number of nuclei and is therefore highly reliable.

*9)*
Figure 7
*is equally difficult to interpret; I suggest eliminating it and merging*
Figure 7
*A, B and D with the Southern blot in*
Figure 4.

*In the images in this figure most of the hTR signal does not co-localize with the Rap1. (Why was Rap1 used here when the previous figure used TRF2?) Was this quantitation carries out in a blinded fashion? If I understand the numbers in white on the final column, cells expressing K78E hTERT and WT TPP1 show 49/52 or 94% co-localization, whilst cells expressing K78E hTERT and E215K TPP1 show 20/51 or 40% co-localization. Isn't this the opposite of what you would expect? This experiment needs to be redone in a blinded fashion, or simply eliminate these co-localization experiments that are problematic*.

We acknowledge the reviewer’s concerns regarding the FISH experiment in Figure 7. Since it is not essential to the conclusions of the paper, we have therefore eliminated it.

*10) The model of TPP1 interaction with TERT is shown in*
Figure 8
*and also in*
Figure 5*. It is not clear that it needs to be in both places. The way that TPP1 is drawn here however might confuse people. The three domains should not have black outlines around them; it makes it look like this is three Different proteins. The color difference between TPP1 and TIN2 is subtle, so hard to know which protein the reddish colors circle belongs to. Eliminating the black line will help this. Similarly if you eliminate the black outline around the TEN domains where it touches the TERT, it will look much more like a domain and not a separate protein*.

The schematic in Figure 5 explains the expected results and logic for a charge-swap experiment. In contrast, Figure 8 depicts a model for telomerase recruitment in cells, and the figures are not redundant. We have modified the colors and black outlines in Figure 8 to enhance the clarity of our model.

*[Editors' note: further clarifications were requested prior to acceptance, as described below*.*]*

*[… It] would be best to simply narrow the conclusion to what can clearly be established from this data: that the TEN domain mutants fail to elongate telomeres like the Wt TERT does*.

Using precise language is always good, so we thank the reviewer for this suggestion. We now state that telomerase with mutations in the TEN-domain fail to elongate telomeres.

*My earlier comment that the double mutant has less activity than the WT TERT is still a concern. This decreased activity is not shown in this figure but is shown in*
Figure 1
*where more quantitative activity assays are done*.

We have added a clause in the text acknowledging that the double mutant fails to elongate telomeres, but we cannot rule out the impact of reduced activity for this mutant. Thus, our conclusion rests on the two single mutants.

*The second point is that the hTR localization to telomeres is expected to be a very difficult thing to measure. It is clear that hTR localization to cajal bodies can be established, and it is expected that many hTR molecules would be localized in this processing center, however it has not been clearly established that a single telomerase at a telomere can be detected by this technique. […] If they authors conclude there may be multiple hTR molecules in these foci, it would be helpful to explain why there would be multiple telomerase molecules and how that affects the charge swap interaction they are proposing as shown in*
Figure 8.

*The uncertainties in these experiments weaken the manuscript. The conclusion on processivity can be made without this FISH data*.

We have now removed the FISH experiments as suggested.

*One final note it was surprising to see the authors state in the rebuttal that under some salt conditions TERT alone is more processive than TERT + Pot1/Tpp1*.

Not true – this is a misunderstanding (see below).

*They say “*Figure 5—figure supplement 1
*is labeled correctly. The gel was a screen carried out under non-physiological salt conditions, and under these conditions telomerase alone is more processive.” This reduced my confidence in the biological result of the Pot1/Tpp1 stimulation. Why is the effect so sensitive to salt conditions*?

The effect of POT1-TPP1 is not so sensitive to salt conditions; it is the intrinsic processivity that is sensitive. The charge swap effect occurs in a variety of salt concentrations, as we stated in the last response. Interestingly, the effect of salt on processivity (higher salt encourages product dissociation and therefore low processivity) has been seen for many different polymerases (e.g., P. Von Hippel et al. (1994) “On the Processivity of Polymerases,” Annals of the N.Y. Academy of Science); that paper also defines physiological salt.

*It is not clear what “non-physiological salt conditions” is and why they would have used these conditions in the experiment shown in*
Figure 5—figure supplement 1*. Presenting this figure will be very confusing to the readers and it is not clear why it is needed*.

The resubmitted manuscript now has this supplemental figure removed, as the reviewer suggests.